# Broadly cross-reactive human antibodies that inhibit genogroup I and II noroviruses

Gabriela Alvarado[1,8], Wilhelm Salmen[2,8], Khalil Ettayebi[2], Liya Hu[3], Banumathi Sankaran[4], Mary K. Estes[2,5], B. V. Venkataram Prasad[2,3,9✉] & James E. Crowe Jr.[1,6,7,9✉]

The rational development of norovirus vaccine candidates requires a deep understanding of the antigenic diversity and mechanisms of neutralization of the virus. Here, we isolate and characterize a panel of broadly cross-reactive naturally occurring human monoclonal IgMs, IgAs and IgGs reactive with human norovirus (HuNoV) genogroup I or II (GI or GII). We note three binding patterns and identify monoclonal antibodies (mAbs) that neutralize at least one GI or GII HuNoV strain when using a histo-blood group antigen (HBGA) blocking assay. The HBGA blocking assay and a virus neutralization assay using human intestinal enteroids reveal that the GII-specific mAb NORO-320, mediates HBGA blocking and neutralization of multiple GII genotypes. The Fab form of NORO-320 neutralizes GII.4 infection more potently than the mAb, however, does not block HBGA binding. The crystal structure of NORO-320 Fab in complex with GII.4 P-domain shows that the antibody recognizes a highly conserved region in the P-domain distant from the HBGA binding site. Dynamic light scattering analysis of GII.4 virus-like particles with mAb NORO-320 shows severe aggregation, suggesting neutralization is by steric hindrance caused by multivalent cross-linking. Aggregation was not observed with the Fab form of NORO-320, suggesting that this clone also has additional inhibitory features.

[1] Department of Pathology, Microbiology and Immunology, Vanderbilt University Medical Center, Nashville, TN, USA. [2] Department of Molecular Virology and Microbiology, Baylor College of Medicine, Houston, TX, USA. [3] The Verna and Marrs McLean Department of Biochemistry and Molecular Biology, Baylor College of Medicine, Houston, TX, USA. [4] Berkeley Center for Structural Biology, Molecular Biophysics and Integrated Bioimaging, Lawrence Berkeley Laboratory, Berkeley, CA, USA. [5] Department of Medicine-Gastroenterology and Hepatology, Baylor College of Medicine, Houston, TX, USA. [6] The Vanderbilt Vaccine Center, Vanderbilt University Medical Center, Nashville, TN, USA. [7] Department of Pediatrics, Vanderbilt University Medical Center, Nashville, TN, USA. [8]These authors contributed equally: Gabriela Alvarado, Wilhelm Salmen [9]These authors jointly supervised this work: B. V. Venkataram Prasad, James E. Crowe, Jr. ✉email: vprasad@bcm.edu; james.crowe@vumc.org

In 2010, there were 1.8 billion cases of diarrheal disease worldwide, and about 18% of these were due to human norovirus (HuNoVs)[1]. In the United States, the Center for Disease Control estimates that there are approximately 19 to 21 million annual cases of acute gastroenteritis caused by HuNoVs[2]. Globally, the economic burden resulting from both direct health system costs and societal costs is estimated to be over $60 billion per year[3]. Thus, there is a substantial global disease burden caused by HuNoVs and a need for sensitive, accurate diagnostics and efficacious therapeutics and vaccines. Progress has been made towards the development of a HuNoV vaccine, with several vaccine candidates currently in clinical trials, but it is unclear whether or not a successful vaccine would need to be reformulated regularly due to the periodic emergence of novel pandemic HuNoV variants.

Noroviruses, comprising a genus within the *Caliciviridae* family, are non-enveloped single-stranded positive-sense RNA viruses. The norovirus genome is organized into 3 open reading frames with the first encoding non-structural proteins, the second encoding the major structural protein (VP1), and the third encoding the minor structural protein (VP2)[4]. VP1 can be divided further into a highly conserved shell (S) domain and a more variable protruding (P) domain[5]. Recombinant expression of the VP1 sequence in insect cells results in the spontaneous formation and release of virus-like particles that are antigenically and morphologically similar to HuNoV virions[6,7]. The amino acid sequence of the major structural protein is also used to classify noroviruses into 10 different genogroups[8]. The HuNoVs can be further subdivided into 49 genotypes[8]. Between genogroups, the genomic nucleotide sequence of structural proteins can differ by more than 50%[9]. Genetic diversity in structural proteins also causes changes in antigenic properties, so it is imperative that we understand the HuNoV-mediated human immunological response to infection and the antigenic variation among circulating strains of HuNoV to develop an effective vaccine.

The intricacies of human antibody-mediated HuNoV cross-reactivity and neutralization remain to be fully elucidated. Numerous studies have assessed the presence of a human polyclonal immune response to HuNoV[10–14]. In this study, we identified and characterized a panel of broadly binding and neutralizing human IgM, IgG, and IgA antibodies from subjects who were infected previously with GII.4 Sydney 2012 HuNoV. Within this panel, we also included 5 of 25 GII.4-reactive mAbs from a previous report: NORO-202A,−232A.2,−279A,−310A, and −320[15]; mAbs from our previous studies of human GI.1-reactive B cell responses did not exhibit cross-reactivity[16] and are not included in this report. Among this panel, we also identified 3 distinct major modes of binding. Initial antigenic mapping studies using strain-specific P and S-domain binding suggested that some of the most cross-reactive mAbs bind to the S-domain[17–19]. X-ray crystallography studies of a broadly reactive GII IgA Ab in complex with GII.4 P-domain suggest that the broad blocking activity for diverse GII strains that we observed is mediated through the steric hindrance of binding to host glycans by recognizing a highly conserved epitope within the P1 and P2 subdomains.

Previous studies have reported the isolation of broadly cross-reactive murine Abs, but none of them blocked VLPs from binding to glycans in vitro, and inhibition of replication of infectious virus using these Abs has not been determined[18,20]. Alpaca Abs with broad blocking activity also have been isolated[21]. More recently, a neutralizing murine Ab was isolated with binding reactivity across selected GII.4 variants[22]. Because of the distinct immunogenetics of diverse species, however, murine or alpaca Ab responses to HuNoV offer little information about the molecular and genetic basis for the human B cell response to infection. Therefore, we set out to isolate HuNoV-specific human mAbs with histo-blood group antigen blocking activity that cross-react with diverse strains and then to map their epitopes. The information we gathered will be useful to inform a rational basis for reformulating HuNoV vaccine candidates since the goal of a vaccine is to elicit a protective response against more than one circulating strain of HuNoV. These studies identified human mAbs of various isotypes with an unexpected degree of breadth and neutralization activity.

## Results

**Isolation of broadly binding anti-HuNoV human mAbs.** To isolate cross-reactive HuNoV human mAbs, we used EBV and additional B cell stimuli to transform memory B cells in PBMCs obtained from patients who were overall healthy but with a previous history of acute gastroenteritis, as previously described[16]. A week later, transformed PBMCs supernatants were tested by ELISA to screen for the expression of mAbs that bound to more than one representative strain HuNoV VLP. The VLPs used to screen were HuNoV GI.1, GI.2, GI.3, GII.3, GII.4, GII.6, GII.13 or GII.17. Each VLP was coated individually and blocked on a microtiter plate before the screening. The bound antibodies were detected using alkaline phosphatase-conjugated goat anti-human κ or λ chain secondary antibodies to capture binding activity by any antibody isotype. Wells that contained transformed B cells expressing mAbs that recognized more than one VLP then were expanded. B cells secreting anti-NoV mAbs were rescued by hybridoma formation. Binding reactivity to pandemic GII.4 Sydney 2012 was previously characterized for 5 of the mAbs included in our panel[15]. The heavy and light variable gene regions were sequenced for all 12 mAbs and the V, D, J, and other variable gene sequence features were analyzed[23] (Supplemental Table 1). Each of the mAbs had unique variable gene sequences, suggesting that cross-reactivity was not limited to one antibody clonotype.

**Binding and blocking activity of cross-reactive mAbs to HuNoV GI and GII VLPs.** To assess the binding reactivity and blocking function of the 12 mAbs, we used indirect ELISA and a VLP blocking assay. The concentration of each mAb was normalized first for the number of antigen-binding sites. We then tested binding starting at a concentration of 500 nM, followed by 11 serial dilutions. We used these data to determine the $EC_{50}$ value of each mAb when binding to HuNoV GI.1, GI.2, GI.3, GII.3, GII.4, GII.6, GII.13 or GII.17 VLPs (Fig. 1A, B). Binding reactivity revealed three distinct binding patterns. NORO-168.2, −156.3, and −170.5, all IgMs, each exhibited wide breadth by binding to all VLPs tested. Both IgAs, NORO-232A.2 and −320 as well as two IgGs, NORO-167.3 and −202A.1 exhibited specificity of binding only for GII variants. The remaining mAbs, NORO-155.5, −178.6, −279A, −310A, and −323A reacted with at least one GI and one GII strain. These binding patterns from natural infection also have been reported recently to follow similar trends in HuNoV vaccination trials[13]. These studies suggest that typical adult human B cell responses to HuNoV antigens include clonotypes encoding both broadly reactive non-neutralizing antibodies and more narrowly reactive neutralizing antibodies.

To determine if any of the isolated cross-reactive mAbs had functional activity, we used a surrogate system to analyze neutralization using porcine gastric mucin (PGM)[24] as described in the Methods section (Fig. 2A, B). We first tested if GI.1, GI.2, GI.3, GII.3, GII.4, GII.6, GII.13, or GII.17 VLPs could bind to the glycans present in PGM and found that GI.3, GII.4, GII.6, and GII.17 VLPs bound to PGM. Therefore, we tested inhibition of

A.

| Isotype | NORO- | κ / λ | EC$_{50}$ (nM) for binding to indicated VLP | | | | | | | |
|---------|-------|-------|---------|---------|---------|---------|---------|---------|---------|---------|
| | | | Genogroup I | | | Genogroup II | | | | |
| | | | GI.1 | GI.2 | GI.3 | GII.3 | GII.4 | GII.6 | GII.13 | GII.17 |
| IgM | 155.5 | κ | > | 415 | > | > | 332 | > | > | > |
| | 156.3 | κ | 42 | 175 | 47 | 46 | 106 | 39 | 54 | 90 |
| | 168.2 | κ | 11 | 265 | 8 | 224 | 285 | 198 | 107 | 153 |
| | 170.5 | κ | 104 | 209 | 90 | 233 | 366 | 474 | 154 | 158 |
| IgG | 167.3 | λ | > | > | > | 115 | 152 | 176 | 383 | 226 |
| | 178.6 | λ | > | > | 83 | 29 | 49 | 20 | 18 | 16 |
| | 202A.1 | λ | > | > | > | 41 | 6 | 6 | 4 | 6 |
| | 279A | λ | > | > | 70 | 14 | 5 | 14 | 10 | 18 |
| | 310A | κ | > | > | 24 | 15 | 72 | 44 | 53 | 40 |
| | 323A | λ | > | > | 3 | > | 146 | > | > | > |
| IgA | 232A.2 | κ | > | > | > | 1 | 1 | 1 | 2 | 1 |
| | 320 | κ | > | > | > | 4 | 2 | 3 | 12 | 3 |

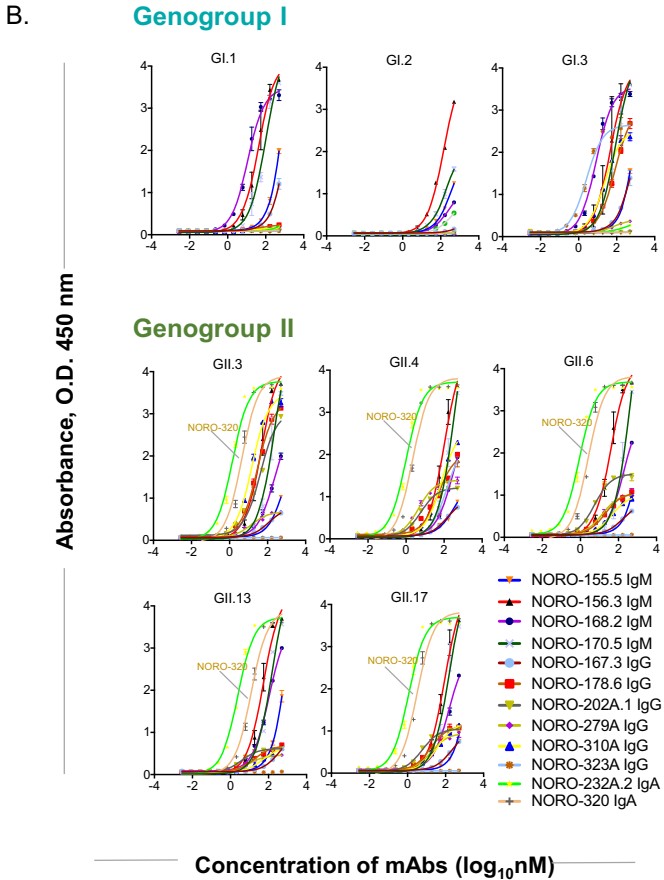

B.

**Genogroup I**

**Genogroup II**

Absorbance, O.D. 450 nm

Concentration of mAbs (log$_{10}$nM)

- NORO-155.5 IgM
- NORO-156.3 IgM
- NORO-168.2 IgM
- NORO-170.5 IgM
- NORO-167.3 IgG
- NORO-178.6 IgG
- NORO-202A.1 IgG
- NORO-279A IgG
- NORO-310A IgG
- NORO-323A IgG
- NORO-232A.2 IgA
- NORO-320 IgA

**Fig. 1 Binding activity of cross-reactive human mAbs to GI and GII VLPs.** An indirect ELISA was used to assess the binding activity of 12 human mAbs to GI.1, GI.2, GI.3, GII.3, GII.4, GII.6, GII.13 or GII.17 VLPs. **A** Half-maximal effective concentration (EC$_{50}$) for binding to VLPs of the indicated genotype. Listed are the isotype, light chain, and EC$_{50}$ value to GI.1, GI.2, GI.3, GII.3, GII.4, GII.6, GII.13, or GII.17 VLPs. The > symbol indicates binding was not detected at the highest concentration tested, 500 nM. Greater EC$_{50}$ values are in the lightest shade of orange and lowest EC$_{50}$ values are in the darkest shade of orange. **B** Representative ELISA binding curves are shown for indicated genotype. The binding curve for NORO-320 IgA, which is studied in detail here, is highlighted. Data presented are means ± SE, $n = 2$ independent study replicates and from one of two independent experiments with similar results. Source Data are provided.

binding of GI.3, GII.4, GII.6, and GII.17 VLPs to PGM. NORO-155.5 and −170.5, both IgMs, and −167.3, and IgG, did not inhibit any of the VLPs tested from binding to PGM in vitro. 9 of the 12 mAbs blocked at least 1 of the 4 VLPs tested. None of the 8 mAbs with binding reactivity to GII.17 VLPs had any strong blocking activity with GII.17; two clones exhibited some activity, but the EC$_{50}$ values were estimated to be >1000 nM. NORO-320, an IgA that bound broadly across selected GII strains, also

blocked GII.4 and GII.6 VLPs from binding to PGM but not GII.17. The absence of GII.17 blocking may stem from a difference in glycans in the pig gastric mucin we used compared to the native human cellular glycans to which GII.17 viruses bind. Another possibility is that the HBGA-binding site remains available even when NORO-320 IgA mediates particle aggregation or disassembly, as suggested by results using dynamic light scattering (see below).

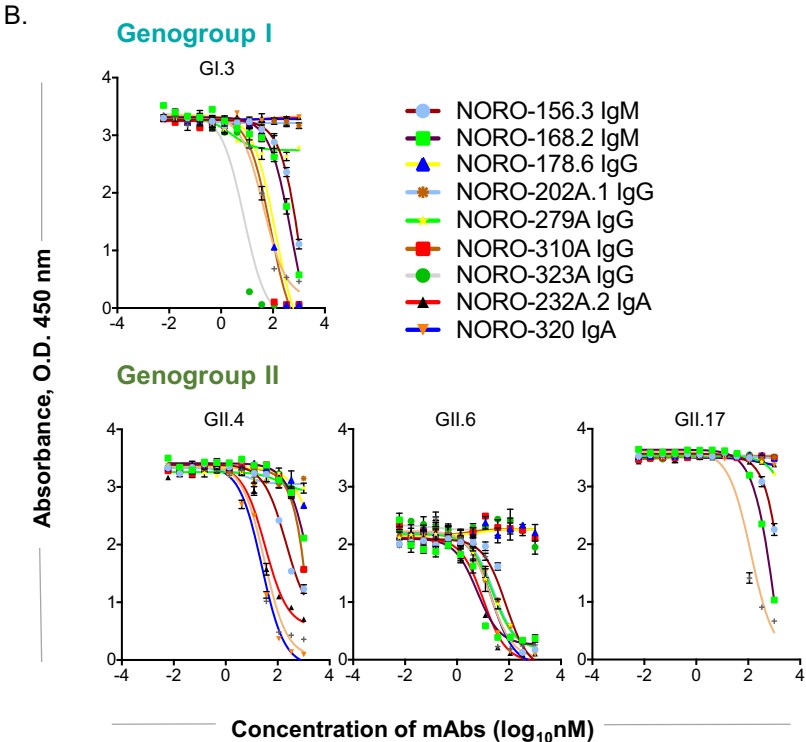

A.

| Isotype | NORO- | EC$_{50}$ (nM) for blocking of binding to PGM | | | |
|---|---|---|---|---|---|
| | | Genogroup I | Genogroup II | | |
| | | GI.3 | GII.4 | GII.6 | GII.17 |
| IgM | 156.3 | 678 | > | 355 | > |
| | 168.2 | > | > | 7 | > |
| IgG | 178.6 | 470 | > | > | > |
| | 202A.2 | > | > | 47 | > |
| | 279A | > | > | 32 | > |
| | 310A | 192 | > | > | > |
| | 323A | 15 | > | > | > |
| IgA | 232A.2 | > | 115 | 20 | > |
| | 320 | > | 73 | 36 | > |

B.

**Genogroup I**

GI.3

- NORO-156.3 IgM
- NORO-168.2 IgM
- NORO-178.6 IgG
- NORO-202A.1 IgG
- NORO-279A IgG
- NORO-310A IgG
- NORO-323A IgG
- NORO-232A.2 IgA
- NORO-320 IgA

**Genogroup II**

GII.4    GII.6    GII.17

Absorbance, O.D. 450 nm

Concentration of mAbs (log$_{10}$nM)

**Fig. 2 Blocking activity of cross-reactive human mAbs for GI or GII VLPs.** Blocking of VLP binding to PGM was used as a surrogate system to test neutralization of GI.3, GII.4, GII.6, or GII.17 VLPs using the indicated human mAbs. **A** Half-maximal effective concentrations (EC$_{50}$) for cross-reactive mAbs when blocking GI or GII VLPs from binding to PGM. EC$_{50}$ values were calculated using a sigmoidal dose-response nonlinear regression analysis after log transformation of the mAb concentrations using GraphPad Prism v 7.0 software. The > symbol indicates the blocking EC$_{50}$ value was greater than 1000 nM. **B** Blocking activity was tested using serial dilutions of each mAb.

**Binding to GI.3, GII.4, GII.6, or GII.17 variant protruding (P) vs shell (S) domains.** The major capsid protein, VP1, which forms the icosahedral capsid, is divided into the protruding (P) and shell (S) domains[25]. To map where the cross-reactive mAbs bind, we first expressed and purified recombinant proteins for GI.3, GII.4, GII.6, and GII.17 HuNoV strains using P-domains expressed in *Escherichia coli* BL-21 cells and S-domains expressed in Sf9 insect cells. Antibody binding to S, P, and VLPs, containing both S and P-domains, was tested and compared. P and S-domain recombinant proteins were coated at equal concentrations of 2 μg/mL on microtiter plates and blocked with 5% nonfat dry milk in 1X DPBS with 0.05% Tween-20. Before adding the mAbs to plates, each mAb was normalized according to the number of antigen-binding sites. We tested binding starting at a concentration of 500 nM followed by 11 serial dilutions of each mAb to obtain the half-maximal binding concentrations. Both IgAs, NORO-232A.2 and −320, appeared to bind specifically to the GII.4, GII.6, and GII.17 P-domains (Fig. 3). We observed wide

ranges of EC$_{50}$ values (2 to 390 nM for P-domains and 3 to 385 nM for S-domains). NORO-320 had some of the lowest EC$_{50}$ values of all the mAbs screened, with EC$_{50}$ values of 2 or 3 nM for each of the P-domains tested. Similarly, previously isolated cross-reactive murine mAbs also have been mapped to the NoV P-domain[26]. NORO-168.2 bound to both the P and S-domains of GI.3, GII.4, GII.6, and GII.17, but in all instances had a lower EC$_{50}$ value when bound to the S-domain. Some mAbs like NORO-155.5 and −156.3 did not bind to any of the P or S-domains tested. Loss of binding may suggest that these mAbs require both the S- and P-domain to be present for Ab binding.

**HBGA-blocking and neutralization of HuNoV infection by NORO-320 mAb and Fab.** In previous studies, we determined that NORO-320 inhibits GII.4 Sydney 2012 virus replication when using a human intestinal enteroid culture[15]. To determine if NORO-320 neutralizes GII.4 because it sterically hinders HuNoV binding to glycans, we tested if blocking activity was influenced

| Isotype | NORO- | Domain specificity | EC50 (nM) for binding to protruding or shell domain protein | | | | | | | |
| | | | Genogroup I | | Genogroup II | | | | | |
| | | | GI.3 | | GII.4 | | GII.6 | | GII.17 | |
| | | | Protruding | Shell | Protruding | Shell | Protruding | Shell | Protruding | Shell |
| IgM | 155.5 | ND | > | > | > | > | > | > | > | > |
| | 156.3 | ND | > | > | > | > | > | > | > | > |
| | 168.2 | P/S | 143 | 37 | 238 | 78 | 135 | 162 | 390 | 183 |
| | 170.5 | S | > | 385 | > | > | > | 304 | > | 255 |
| IgG | 167.3 | S | > | > | > | 90 | > | > | > | > |
| | 178.6 | P/S | 336 | > | > | 8 | > | > | > | > |
| | 202A.2 | S | > | > | > | 3 | > | > | > | > |
| | 279A | S | > | > | > | 3 | > | > | > | > |
| | 310A | P/S | 60 | > | 219 | 19 | > | > | > | > |
| | 323A | P | 7 | > | 172 | > | > | > | > | > |
| IgA | 232A.2 | P | > | > | 3 | > | 2 | > | 2 | > |
| | 320 | P | > | > | 2 | > | 3 | > | 3 | > |

**Fig. 3 Half-maximal effective concentrations (EC50) for binding of 12 cross-reactive human mAbs to protruding or shell domain.** GI.3, GII.4, GII.6, or GII.17 HuNoV strain protruding or shell domain proteins were used as antigen in an indirect ELISA. The > symbol indicates binding was not detected at the highest concentration tested, 500 nM. ND, not determined; P, protruding domain; S, shell domain; P/S, protruding and shell domain.

by the molecular weight or size of NORO-320, using recombinant Fab, IgG, or IgA isotypes of NORO-320. To verify the molecular weight of the original hybridoma-expressed IgA and each of the recombinantly expressed mAbs, 4 μg of each mAb, along with a set of control mAbs of known molecular weight, were resolved on an SDS-PAGE gel under non-reducing conditions (Supplemental Fig. 1). All the mAbs were of the expected apparent molecular weight, dIgA ~350 kDa, IgG ~150 kDa, and Fab ~50 kDa. We then tested NORO-320 IgA, recombinant IgG, recombinant Fab, and an irrelevant mAb as a control for their ability to inhibit GII.4 Sydney 2012 VLPs from binding to PGM in vitro. As hypothesized, blocking activity varied by the form of the antibody. The large NORO-320 IgA had the lowest EC50 value followed by NORO-320 recombinant IgG (Fig. 4). Recombinant NORO-320 Fab did not block GII.4 Sydney 2012 VLPs from binding to PGM at concentrations as high as 1,000 nM. This finding indicates that the dimeric NORO-320 likely neutralizes GII strains broadly because of the capacity of this large molecule to mediate steric hindrance to receptor binding by cross-linking and aggregating viral particles (Supplemental Fig. 2).

Furthermore, to examine if the NORO-320 Fab also lacks the ability to neutralize the infectious virus, we performed neutralization assays using the enteroid culture system[27]. Remarkably, we observed that NORO-320 Fab mediated neutralization of viral replication for both GII.4 and GII.17 HuNoVs (Fig. 5). While NORO-320 IgA neutralizes GII.4 with a high IC50 of 11,690 ng/mL, the NORO-320 Fab exhibits an IC50 of 2,950 ng/mL.

**Crystal structure of NORO-320 Fab in complex with GII.4 P-domain.** To understand how NORO-320 binds so broadly and neutralizes diverse GII HuNoV strains without blocking glycan binding, we determined the crystal structure of the NORO-320 Fab in complex with the GII.4 P-domain at a resolution of 2.3 Å (Figs. 6A, B). According to the structure, NORO-320 Fab binds perpendicular to the 2-fold axis of the P-domain dimer near a region close to the shell domain and significantly distant from the HBGA-binding site to inhibit GII.4 VLP-carbohydrate binding. The superimposition of the structures of GII.4 P-domain in complex with NORO-320 and in complex with HBGA revealed

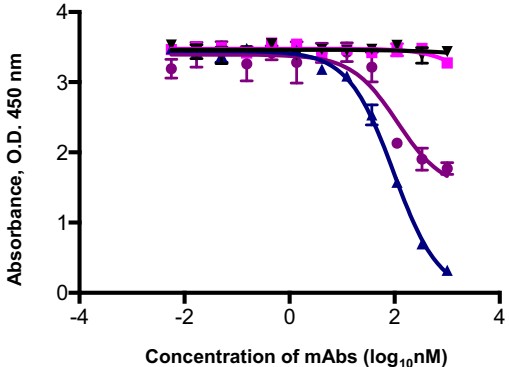

| NORO-320 | EC50 (nM) |
| --- | --- |
| dIgA | 96 |
| rIgG | 113 |
| rFab | > |

- NORO-320 rIgG
- NORO-320 rFab
- NORO-320 IgA
- No mAb

**Fig. 4 HBGA blocking of GII.4 VLPs by NORO-320 is a result of steric hindrance.** NORO-320 was expressed recombinantly as Fab (rFab) or IgG (rIgG) forms and purified. GII.4 VLPs were preincubated with either NORO-320 rFab, rIgG or the original hybridoma-secreted (Hyb) dimeric IgA and added to wells that had been coated previously with PGM. Half-maximal concentrations (EC50) for the three antibodies tested are listed. The > symbol indicates blocking EC50 value was greater than 1000 Mn. Data presented are means ± SE, n = 2 independent study replicates and from one of two independent experiments with similar results. Source Data are provided.

that the P-domain dimer structure remains invariant, with an r. m.s.d. of ~1.1 Å for the matching Cα atoms. These structural observations indicate that the Fab form of NORO-320 cannot affect glycan binding either directly or allosterically. These observations are consistent with our findings from the HBGA-blocking assays showing that NORO-320 Fab does not inhibit glycan binding.

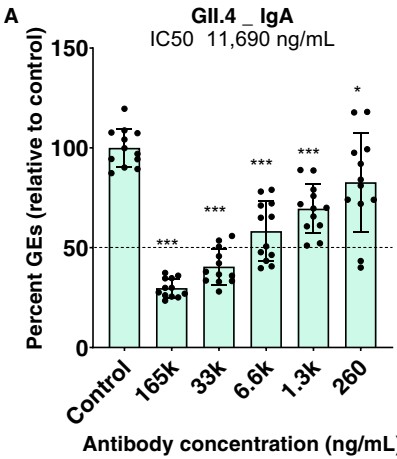

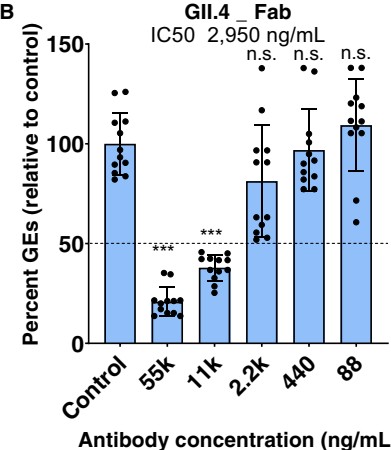

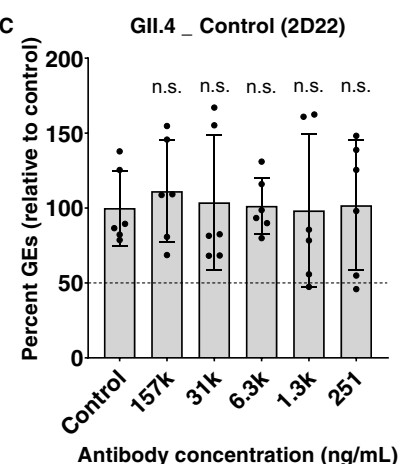

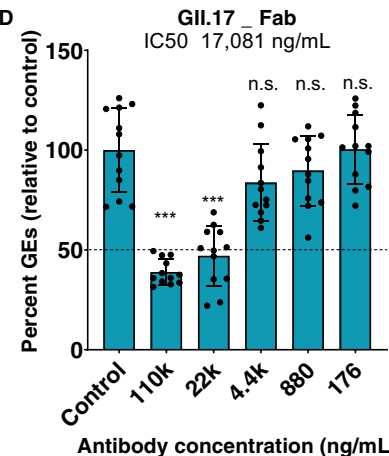

**Fig. 5 Neutralization of GII.4 and GII.17 in human intestinal enteroid system.** HuNoV was mixed with an equal volume of medium or dilutions of the indicated antibody, and then incubated at 37 °C for 1 h. Human intestinal enteroid monolayers were inoculated with each virus-antibody mixture for 1 h at 37 °C in the presence of 500 μM GCDCA. The monolayers were washed twice and then cultured in the presence of GCDCA for 24 h. Compiled data from two experiments are presented. Each data bar represents the mean ± SD of six wells of inoculated HIE monolayers. Error bars denote standard deviation and individual points are shown (n = 12 for IgA and Fab and n = 6 for GII.4_control). Percent reduction in viral genome equivalents (GEs) relative to medium (100%) was determined. The dotted line represents 50% neutralization. Significance relative to the control was determined using two-tailed Student's t test. Exact ρ values from left to right: (**A**) ***ρ = 1.10E−16, 2.30E−13, 4.78E−08, 9.70E−07, *ρ = 0.03; (**B**) ***ρ = 1.3E−13, 1.13E−11; (**C**) all differences are n.s., not significant; (**D**) ***ρ = 2.45E−09, 4.13E−07.

**Molecular details of recognition of GII.4 P-domain by NORO-320.** The crystal structure of NORO-320 Fab-GII.4 P-domain complex shows that the antibody makes extensive interactions with the P-domain. The Fab binding site on the P-domain is formed by residues from the P1 subdomain of one subunit and the P2′ subdomain of its dimeric partner (Figs. 6C, D). The paratope in NORO-320 includes residues from the CDRs of both light and heavy chains. The Fab binding is stabilized by both hydrogen bond and hydrophobic interactions (Fig. 7A). For instance, the sidechain of N479 in the P1 subdomain hydrogen bonds with I54 from CDRH2 and E74 of a non-CDR loop, whereas residues L486, V508, P510, P511, and N512 are involved in hydrophobic interactions with residues from CDRH2 and CDRH3 (Fig. 7B). Residue D312 of the P2′ subdomain interacts with K119 and Y120 of CDRH3, involving both direct hydrogen bond and hydrophobic interactions (Fig. 7C). In addition to CDRHs, three light chain residues Y35 and Y37 of CDRL1 and L55 of CDRL2 form hydrophobic interactions with P313′ and T314′ of the P1′ subdomain. To understand how NORO-320 can bind to VLPs of various GII strains, we aligned the P-domain amino acid sequences of GI.1, GII.3, GII.4, GII.6, GII.13, and

GII.17 (Fig. 8). Sequence alignment revealed 78 to 89% conservation at these sites among the GII strains compared to 54 to 59% conservation for the entire P-domain sequence. In contrast, the epitope sequence in the GI genogroup shows significant changes in this region accounting for only 44% sequence similarity. The high level of sequence conservation within the epitope of GII strains provides insight into why NORO-320 binds broadly among GII.3, GII.4, GII.6, GII.13, and GII.17 strains.

**Dynamic light scattering analysis of GII.4 particle integrity by NORO-320.** To investigate the effects of binding of NORO-320 mAb or Fab in the context of GII.4 capsid, we carried out dynamic light scattering studies with GII.4 VLPs (Figs. 9A, B). These studies showed that with GII.4 VLP and the IgA form of NORO-320, a large fraction of the sample has particles of 200–500 nm in diameter. This 200–500 nm peak would correspond to clumping of approximately 4–12 intact viral particles, based on an average VLP diameter of 40 nm. In contrast, when GII.4 VLPs were treated with NORO-320 Fab at a 1:1 or 1:10 ratio of VP1:Fab, we did not observe an alteration in particle diameter (Fig. 9B). GII.4 VLPs were also treated with NORO-320

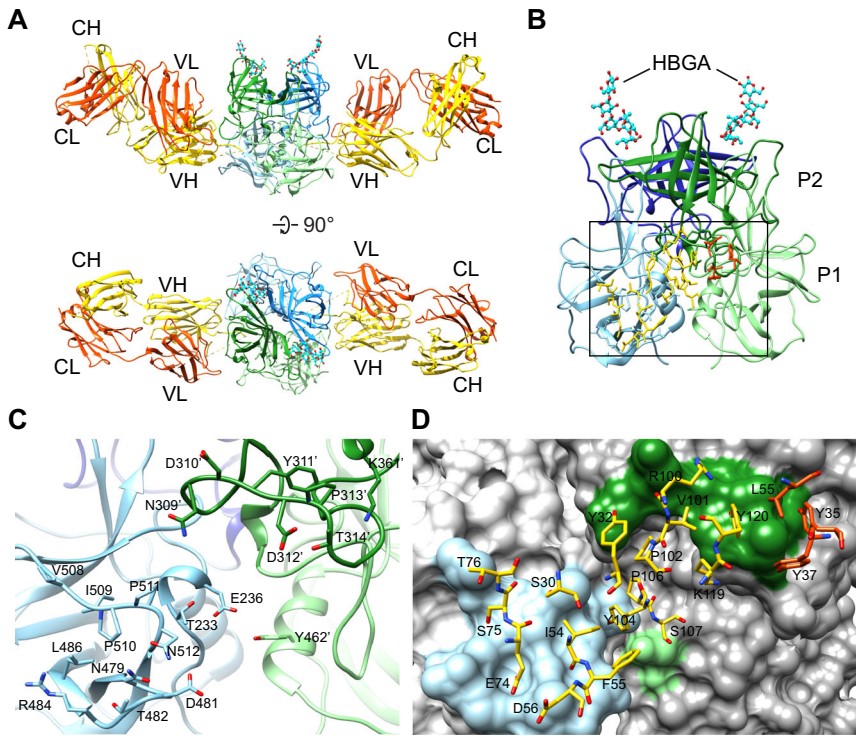

**Fig. 6 NORO-320 Fab in complex with GII.4 P-domain. A** X-ray crystal structure of the NORO-320-GII.4 P-domain complex. The two P-domain subunits in the dimer are colored in blue and green. NORO-320 Fab (yellow-heavy chain and red-light chain) along with two molecules of H-type 1 pentasaccharide (stick model) are modeled to indicate the glycan-binding sites for reference. Depicted are side and top views of the complex. **B** Side view of NORO-320 in complex with GII.4 P-domain showing the interacting Fab residues (stick model) with the light and heavy chain residues shown in red and yellow, respectively, interacting. **C** Close-up view of the Fab binding site (black box in B) showing the P-domain residues (stick models) that interact with the Fab. Residues in the two-fold related subunit (green) are hyphenated. **D** Close-up view of the Fab residues (light chain-red, heavy chain yellow) that interact with the GII.4 P-domain (shown in surface representation).

Fab at pH 6, pH 7, or pH 8 and incubated at 25 °C, 37 °C, or 40 °C for 30 min (Supplemental Fig. 4). We did not observe an increased susceptibility to particle aggregation or disassembly based on pH or temperature.

**Bis-ANS assay to probe local conformational changes**. To further investigate the possibility that binding of NORO-320 Fab could induce local conformational changes in the GII.4 VLP, we used a bis-ANS fluorescence assay. This approach has been used previously to detect possible local conformational changes in feline calicivirus capsid protein upon incubation with the soluble cellular receptor feline junctional adhesion molecule A (fJAM-A)[28]. Upon incubation of GII.4 VLP with NORO-320 IgA or Fab, we did not detect a significant increase in bis-ANS fluorescence at 25 °C or 37 °C (Supplemental Fig. 3). These studies, consistent with the results from the DLS experiments, suggest that the binding of NORO-320 Fab does not cause any significant conformational changes in the VLP.

## Discussion

Isolation of naturally occurring broad-spectrum human mAbs to HuNoV holds great promise for the discovery of new candidate therapeutics, as well as identifying critical epitopes for the rational design of new structure-based broadly protective HuNoV vaccines. In the past, the genetic and antigenic diversity across circulating strains of HuNoV has made the generation of a broadly immunogenic vaccine extremely difficult. The primary goal of this study was to define the molecular and structural determinants of cross-reactivity and neutralization, using human mAbs to circulating strains of HuNoV. Previous studies have

characterized the antigenic landscape of specific HuNoV strains, but with the rapid emergence of new genetically diverse strains, there is a need to map new immunogenic epitopes. This new information builds upon previous studies to help track the evolution of HuNoVs[29]. Identification of antigenic epitopes using human mAbs also will provide insight into the immunogenicity of HuNoVs. Of particular note, the binding patterns of clones in these panels of mAbs isolated from individuals following natural infection are consistent with recently reported data reported in human norovirus vaccination trials. In those studies, investigators identified one class of HuNoV circulating antibodies that exhibit extensive binding breath recognizing GI strains and GII strains but having no blocking activity and identified a second class containing antibodies that exhibit a more narrow range of reactivity but have blocking and neutralization activity[13]. These studies consistently show that human B cell responses to infection or vaccination with HuNoV antigens induce both broadly reactive non-neutralizing antibodies and strain-specific neutralizing antibodies.

Here, we described the isolation of 12 anti-NoV cross-reactive human mAbs, 4 IgMs, 6 IgGs, and 2 IgAs, from subjects with a previous history of acute gastroenteritis. To determine the functional activity of the isolated mAbs, we used a previously validated surrogate blocking assay that measures the inhibition of VLPs from binding to glycans in vitro[30]. Not all of the strains tested bind to the same glycan, so we were only able to test the inhibition of binding for HuNoV VLPs GI.3, GII.4, GII.6, and GII.17. Of the 12 mAbs isolated, 9 mAbs blocked at least 1 of VLPs tested from binding to PGM with $EC_{50}$ values less than 1 µM. It should be noted that the use of a single VLP to represent a genotype is not comprehensive, as significant sequence variation exists within genotypes.

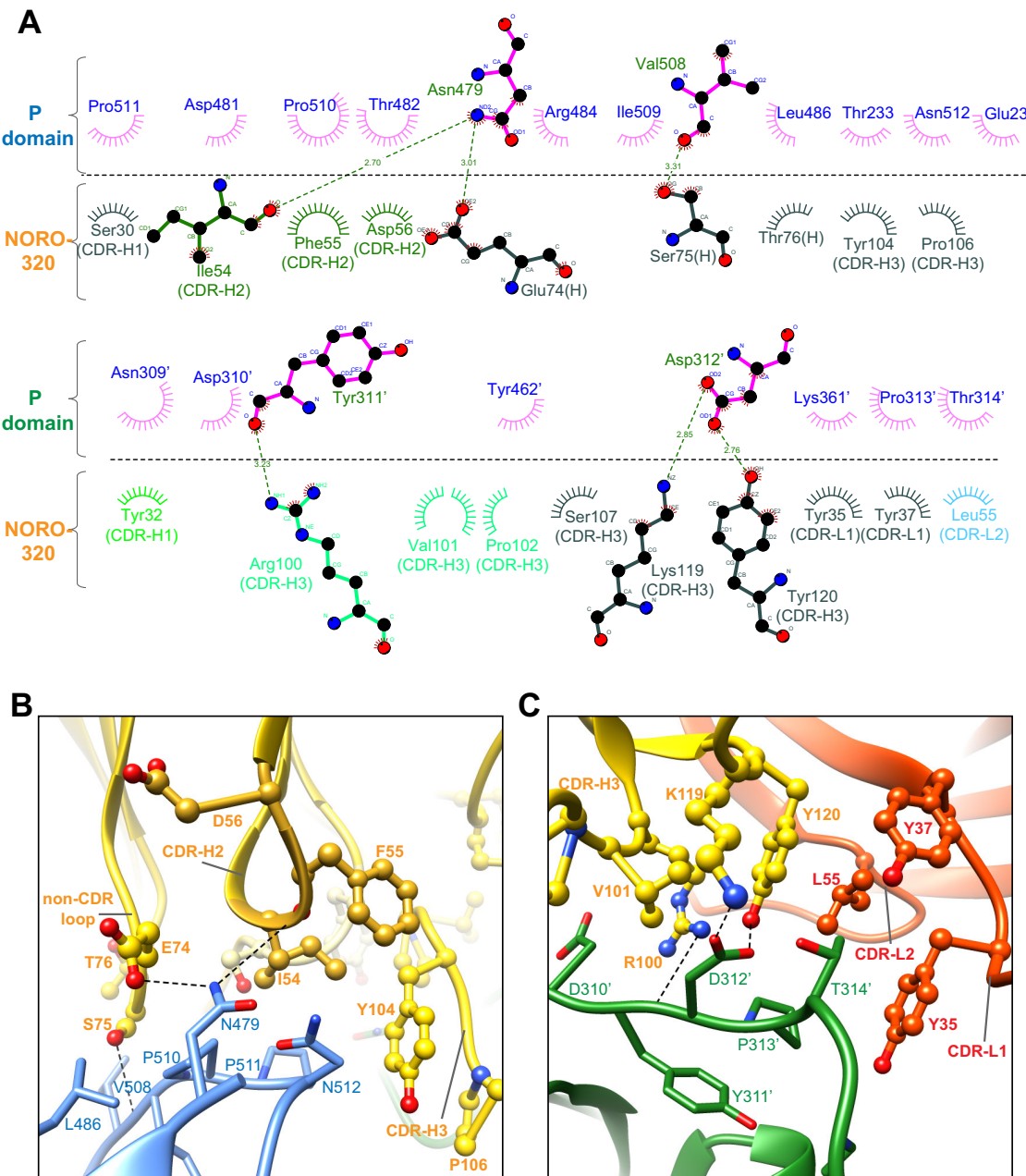

**Fig. 7 Molecular interactions between NORO-320 and GII.4 P-domain. A** Antibody plot analysis using the program LigPlot+ v.2.1[49]. The P2 subdomain residues in the 2-fold related subunit are hyphenated. The hydrogen bonds are shown as green dashed lines, and the hydrophobic contacts are short spokes radiating from each atom or residue. **B**, **C** The interactions of P1 or P2 subdomains with NORO-320. The side chains of mAb and P-domain are represented with ball-and-stick and stick models, respectively, and colored as in Fig. 4. The hydrogen bonds are shown as black dashed lines.

Prior to these studies, we expected that cross-reactive and neutralizing human mAbs would bind predominantly to the P subdomain of the circulating strains, similar to the findings for cross-reactive murine Abs[31]. The P-domain has more surface exposure on a viral particle than the S-domain and should, therefore, be more accessible than the S-domain. Here we describe a crystal structure of a human-derived neutralizing antibody in complex with a GII.4 strain of HuNoV. Both previously characterized mAbs, the human IgA 5I2 and the mouse IgG 10E9, target regions adjacent or at the HBGA-binding site on the P2 subdomain thereby directly preventing the binding of glycans[22,32]. The crystal structure of Nano-85, an alpaca-derived nanobody, which is much smaller in molecular size when compared to a Fab, in complex with the GII.4 P-domain also was reported recently[33]. Nano-85 binds toward the proximal end of the P-domain dimer, which in the context of the capsid would be closer to the S and P-domain interface. The epitope recognized by this nanobody which consists of W520, N522, and T526 is distinct from that recognized by NORO-320. Based on negative-stain images of VLPs in complex with this nanobody, it was suggested that the binding disrupts particle assembly. The crystal structure of the broadly reactive GII.4-blocking human antibody A1431 in complex with the GII.4 P-domain also has been reported recently[13]. Unlike NORO-320, mAb A1431 recognizes an epitope on the P-domain protomer within the P1 and P2 subdomain cleft, primarily recognizing residues Q402, W403, Q504, and D506. In comparison to this GII.4-specific antibody, NORO-320 recognizes a highly conserved epitope that allows it to

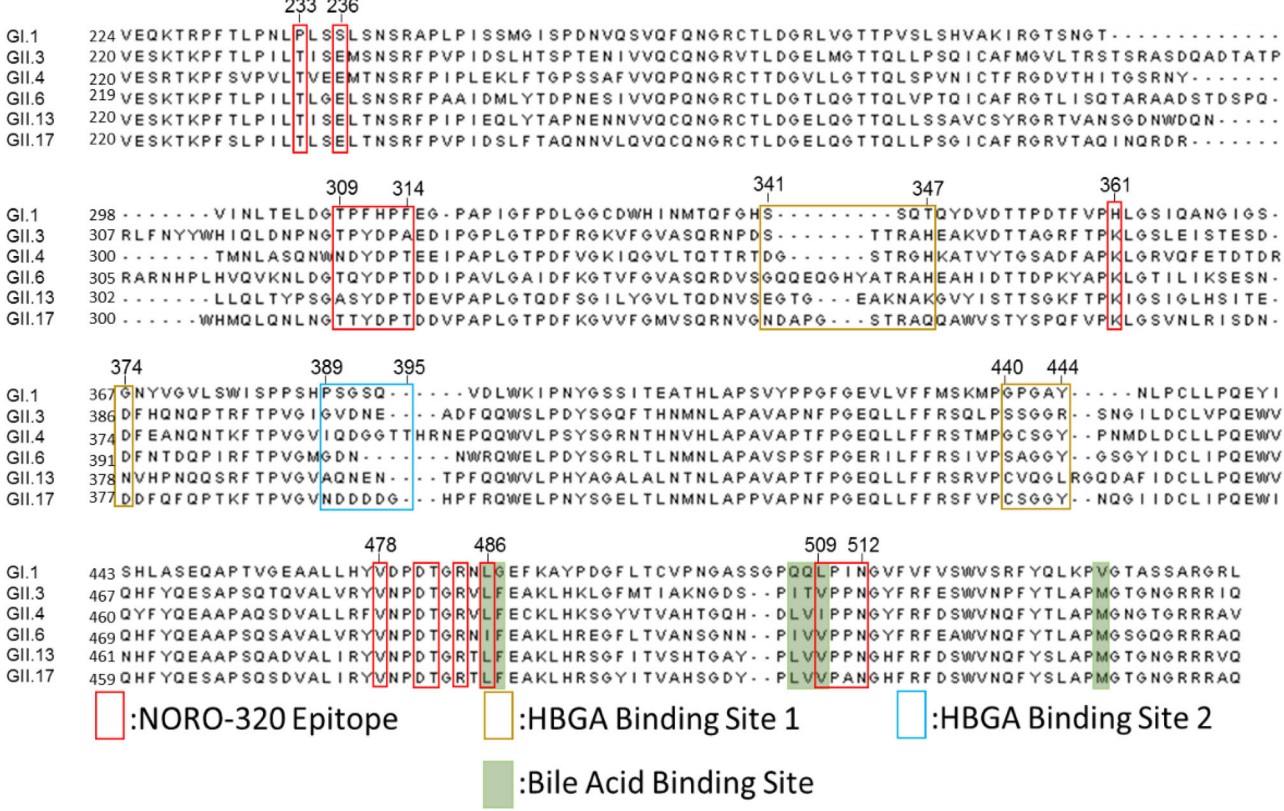

**Fig. 8 Amino acid sequence alignment of the protruding domain of GI.1, GII.3, GII.4, GII.13, and GII.17 strains of HuNoV.** The protruding domain amino acid sequences of GII.3, GII.4, GII.13, and GII.17, the GII strains tested for which NORO-320 exhibited reactivity, and GI.1 are aligned. The 18 residues identified on GII.4 that interact with the highly cross-reactive mAb NORO-320 are boxed in red. The residues previously reported to be involved in GII.4 HBGA binding[55] are boxed in gold and blue. The recently identified low-affinity bile acid-binding site[56] is boxed in green.

recognize not only GII.4 variants, but additional GII genotypes, and it mediates HBGA blocking for GII.6 (Fig. 2A).

It is indeed intriguing that even though NORO-320 does not directly bind in close proximity to the HBGA-binding site, it still inhibits GII.4 and GII.6 VLPs from binding to glycans in vitro and inhibits viral replication of GII.4 Sydney 2012 and GII.17 viruses[15]. Our blocking studies using IgG and Fab recombinant variants, as well as the originally isolated antibody that was a dimeric IgA molecule, show that the molecular size of the mAb influenced the degree of HBGA blocking (Fig. 4). Here, blocking potential increased with the increase in molecular size of the NORO-320 variant tested. Therefore, blocking appears to result from the action of NORO-320 to block GII.4 VLPs sterically from binding to glycans in vitro. The additional Fc region projecting out from the Fab, as in the context of IgA or IgG, may sterically hinder the glycan-binding sites in the neighboring VP1 subunits.

Interestingly, however, when we performed neutralization assays using the previously characterized enteroid culture system, we observed recombinant NORO-320 Fab exhibited similar levels of neutralization in comparison to full-length IgA and mediated neutralization of GII.17 (Fig. 5). Modeling of the Fab binding to the P-domain in the context of the capsid using the only available X-ray structure of the GI.1 indicates that the constant domain of the Fab likely clashes with the neighboring VP1 subunits. This conflict may affect particle disassembly, which may explain in part the mechanism by which NORO-320 Fab neutralizes GII.4 infection (Supplemental Fig. 2). However, the DLS data presented here indicate that binding of NORO-320 Fab does not compromise particle integrity, because the diameter of the VLPs remains the same as that of the VLPs in the absence of the NORO-320 Fab (Fig. 9). In contrast, we observe a significant increase in particle

size from 40 nm to 210 nm with IgA, strongly suggestive of particle aggregation due to multivalent cross-linking, or aggregation of the sub-assemblies following particle disassembly. The results from the bis-ANS fluorescence assay also suggest that the binding of NORO-320 Fab does not cause any significant conformational changes in the VLP (Supplemental Fig. 3). Based on these observations, we suggest that NORO-320 IgA likely mediates neutralization principally by particle aggregation or disassembly of GII.4 particles. Since the Fab form of NORO-320 also mediates neutralization without causing aggregation, there must also be additional inhibitory features of this antibody that we did not define here that contribute to neutralization.

A recent study[34] has reported that GII.3 VLPs exhibit two different conformations of the P-domain dimer in which one conformation of the P-domain is rotated by ~70° and elevated above the shell domain compared to the other conformation. A similar rotated and elevated state also is observed in the case of murine norovirus capsid structure[35]. While the mechanism and impact of this conformational change are not yet clear for HuNoV, for murine norovirus such a conformational change contributes to increased accessibility to its cellular receptor CD300lf and enhancement of infection efficiency[36–40]. The binding of NORO-320 Fab possibly could play a role in impeding such rotation and elevation between conformations, resulting in lower infection efficiency, although we did not demonstrate this effect directly. Such an effect of this Fab could pertain to GII.3, GII.6, and GII.17 variants, since the amino acids critical for the interaction between the GII.4 P-domain and NORO-320 are conserved in those strains.

Taken together, our results presented here suggest that although there is a high degree of sequence and antigenic

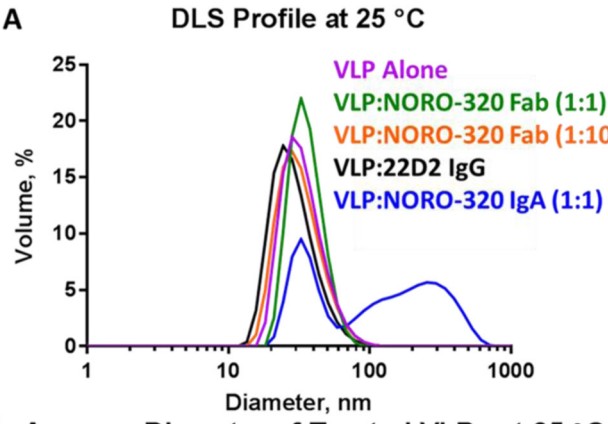

**Fig. 9 Dynamic light scattering of mAb NORO-320 and GII.4 Sydney VLP.** The hydrodynamic diameters of treated or untreated GII.4 HuNoV VLPs were measured using dynamic light scattering (DLS) on a ZetaSizer Nano instrument (Malvern Instruments, UK). **A** Complete dynamic light scattering profile of the four tested conditions: GII.4 HOV VLP alone or in complex with 22D2 IgG, NORO-320 IgA, or Fab at a molar ratio 1:1 or 1:10. A dengue virus-specific antibody, 2D22 IgG, was used as a control at a molar ratio 1:10. **B** Average diameters were calculated for each sample condition using Zetasizer software. Samples were diluted to a final concentration of 330 nM for each component in phosphate-buffered saline pH 6, and a 3,300 nM concentration of NORO-320 Fab was prepared for the condition labeled VLP:NORO-320 Fab (1:10). Individual data points are presented as black or gray symbols. Three × 12 measurement runs were performed with standard settings (refractive index 1.335, viscosity 0.9, temperature 25 °C). Each data bar represents the mean ± SD of $n = 3$ independent experiments.

diversity in the capsid protein VP1 among circulating strains of HuNoVs, common protective antigenic sites exist among these genotypes. Recognition of the P1 subdomain and the more conserved S-domain of VP1 by human mAbs could be the molecular basis for broad cross-reactive neutralization. The neutralizing HuNoV epitope identified also informs us about a critical antigenic GII site that could later be used in the reformulation of broadly protective HuNoV vaccine candidates. These human mAbs also could be directly used as a prophylactic, a therapeutic, or a reagent for diagnosis.

## Methods
**Generation of virus-like particles**. Virus-like particles (VLPs) based on HuNoV strains GI.1 (M87661), GI.2 (AF435807), GI.3 (AF439267), GII.3 (TCH02-104), GII.4 (AFV08795.1), GII.6 (AF414410), GII.13 (JN899242) and GII.17 (AB983218) were expressed recombinantly and purified as previously described (Jiang et al., 1992). We used a baculovirus recombinant protein expression system for VLP production. We cloned the VP1 and VP2, major and minor, protein capsid sequences from each strain into the transfer vector pVL1392 (Epoch Life Science, Inc). Sf9 insect cells were co-transfected with a transfer vector corresponding to a

specific strain and with a bacmid vector. Recombinant baculovirus was isolated and expanded. VLPs were purified from cell culture supernatants using a sucrose and cesium chloride gradient. VLP formation was verified using electron microscopy.

**Reactivity to VLPs by ELISA**. An ELISA was used to testing binding of human mAbs to VLPs, as was previously described[16]. Each VLP was coated individually at 1 μg/mL on 384-well microtiter plates at 4 °C overnight. Plates then were blocked for one hr at room temperature using 5% nonfat dry milk in PBS with 0.05% Tween-20. For screening and EC$_{50}$ analysis, antibody reactivity to VLPs was detected using horseradish peroxidase (HRP) tagged anti-κ or -λ chain secondary antibodies (Southern Biotech). 1-Step$^{TM}$ Ultra-TMB Substrate Solution (Pierce Thermo Fisher) was used to detect HRP activity.

**Human subjects**. The Vanderbilt University Medical Center Institutional Review Board approved the protocol used in this study in which six adult individuals participated. All participants provided written informed consent before we obtained blood samples. The subjects had a previous history of acute gastroenteritis but were otherwise healthy.

**Human hybridoma generation**. Human hybridomas secreting human mAbs were generated as previously described[15]. Briefly, PBMCs were isolated from human subject blood samples using Ficoll-Histopaque and density gradient centrifugation and then cryopreserved. Later, cells were thawed, transformed using Epstein-Barr virus, CpG10103, cyclosporine A and a Chk2 inhibitor and plated in a 384-well plate. Transformed cells were incubated at 37 °C for 7 days, and then expanded into 96-well plates containing irradiated human PBMCs. Four days later, cell supernatants were screened by indirect ELISA for the presence of anti-norovirus VLP cross-reactive mAbs. B cells secreting cross-reactive mAbs were electrofused to HMMA2.5 myeloma cells and plated in medium containing hypoxanthine, aminopterin, thymidine, and ouabain. Hybridoma cell lines were incubated at 37 °C for 14 days, and then supernatants were screened by indirect ELISA for the production of cross-reactive mAbs. Cell lines expressing cross-reactive mAbs then were cloned biologically using single-cell fluorescence-activated cell sorting.

**Purification of cross-reactive mAbs**. After cloning, hybridoma cell lines producing cross-reactive mAbs were expanded gradually from 48-well plates to 12-well plates, T-25, T-75 and eventually to four T-225 flasks for each cell line. Supernatant from each cell line also was screened by ELISA to determine the corresponding light chain for each clone. Following 4 weeks of incubation at 37 °C, supernatant from the four T-225 flasks was harvested and filtered through a 0.4-μm filter. The supernatant was filtered using column chromatography, specifically HiTrap KappaSelect and Lambda FabSelect affinity resins (GE Healthcare Life Sciences). To obtain varying forms of NORO-320, we expressed the heavy and kappa light chain variable domains using Fab or IgG protein recombinant expression vectors. cDNAs encoding the corresponding heavy and light chains were transfected using Expi-CHO$^{TM}$ (Chinese hamster ovary) cells (Thermo Fisher Scientific).

**VLP-carbohydrate mAb blocking assay**. To test the ability of each mAb to inhibit the interaction between the selected VLPs and glycans in vitro, we used a blocking assay. As previously described, we coated microtiter plates with 10 μg/mL of pig gastric mucin Type III (Sigma) for 4 hr at room temperature. Porcine gastric mucin (PGM) purified from porcine stomach mucosa contains both H and Lewis antigens, α-1,2-fucose and α-1,4-fucose[24,41,42]. Plates then were blocked overnight at 4 °C in 5% nonfat dry milk. VLPs at 0.5 μg/mL were pretreated with serially diluted concentrations of each mAb for 1 h at room temperature. The optimal concentrations of mAbs were normalized before testing blocking ability and tested at concentrations beginning at 1000 nM and then diluted serially. VLP-mAb complexes were added to the PGM-coated and blocked microtiter plates. After 1 hr of incubation, the plates were washed three times with PBST and the same was done in between each step. Bound VLPs were tested using murine serum-containing anti-GI.3, GII.4, GII.6, or GII.17 polyclonal antibodies, followed by an HRP conjugated goat anti-mouse IgG human adsorbed antibody. Optical density was measured at 450 nm using a Synergy HT Microplate Reader (BioTek). Blocking studies also were repeated three times.

**Expression and purification of protruding and shell domain for selected HuNoV strains to be used in Ab binding studies**. To map the epitope of cross-reactive mAbs, we first recombinantly expressed P1 and P2 domain sequences or shell domain of GI.3 (AF439267), GII.4 (AFV08795.1), GII.6 (AF414410), GII.13 or GII.17 (AB983218). P-domain sequences were cloned into the pGEX-4T-1 expression vector with a GST tag and thrombin cleavage site. The P-domain then was expressed in *Escherichia coli* BL-21 cells and purified using a Glutathione Sepharose Fast Flow Column (GE Healthcare) and column chromatography. The S-domain sequences were cloned into pVL1392 and co-transfected with a bacmid vector into Sf9 insect cells. Recombinant baculovirus particles then were harvested and used to inoculate Sf9 cells. S-domain particles were then purified from the inoculated Sf9 cell culture supernatant using a sucrose and a cesium chloride cushion gradient.

**Expression, purification, and crystallization of GII.4 P-domain and NORO-320 Fab.** The sequence for the GII.4 protruding domain was cloned into the expression vector pMal-C2E (New England BioLabs). The expression vector includes a N-terminal His$_6$-maltose binding protein (MBP) tag and a tobacco etch virus (TEV) protease cleavage site between the MBP and P-domain sequence. The P-domain was expressed in *E. coli* BL21(DE3) and purified using an AffiPure Ni-NTA agarose bead column (GenDepot). The His-MBP tag was then removed using TEV protease and separated from the P-domain by purifying it once again using His-Trap (GE Healthcare), MBPTrap (GE Healthcare) affinity columns and size exclusion chromatography. Finally, the purified P-domain was concentrated and stored in 20 mM Tris-HCl buffer (pH 7.2) containing 150 mM NaCl, and 2.5 mM MgCl$_2$.

The nucleotide sequences of the variable domain of mAb NORO-320 were optimized for mammalian expression and synthesized (GenScript) for expression and purification of recombinant Fab. The heavy chain fragment was cloned into a vector for the expression of recombinant human Fabs[43]. The light chain was cloned into a vector for κ light chain. Each vector was transformed independently into *E. coli* cells, and DNA then was purified. Both the heavy and light chain encoding vectors were transfected into CHO cells using an ExpiCHO$^{TM}$ expression system. Cell supernatant was collected, centrifuged, and filtered using a 0.45 μm filter. NORO-320 Fab was purified by affinity chromatography using a KappaSelect (GE Healthcare).

Purified GII.4 P-domain and NORO-320 Fab were combined in a 1:1.5 molar ration and incubated for 1 hr at 4 °C. The mixture was passed through an S200pg 16/60 gel filtration column, and the peak corresponding to the complex was collected. The size of the complex and the presence of both proteins was validated on an SDS-PAGE gel. The peak fractions then were pooled and concentrated to 10 mg/mL for crystallization trials. Crystallization screening using hanging-drop vapor diffusion method at 20 °C was set up using a Mosquito nanoliter handling system (TTP LabTech) with commercially available crystal screens, and crystals were visualized by using a Rock Imager (Formulatrix). The GII.4 P-domain–NORO-320 Fab complex crystallized in a buffer containing 0.1 M BIS-TRIS prop 8.5 pH, 0.2 M KSCN, 20% w/v PEG 3350. Crystals diffracted to 2.25 Å resolution.

**Diffraction, data collection, and structure determination.** Diffraction data were collected on beamline 5.0.1 at Advanced Light Source (Berkeley, CA). Diffraction data were processed using HKL2000[44]. The previously published GII.4 (strain TCH05) P-domain structure (PDB ID 3SJP) and the neutralizing Fab 5I2 (PDB ID 5KW9) were used as the search models by molecular replacement using program PHASER[45]. Iterative cycles of refinement and further model building were carried out using PHENIX[46] and COOT programs[47]. During the course of the refinement, and following the final refinement, the stereochemistry of the structures was checked using MolProbity[48]. Data refinement and statistics are given in Table 1. The interactions between P-domain and the Fab for NORO-320 were analyzed using LigPlot+ v.2.1[49]. Figures were prepared using Chimera[50]. Sequence alignment and sequence comparisons we analyzed using EMBL-EBI multiple sequence alignment software[51].

**Dynamic light scattering.** The hydrodynamic diameters of treated or untreated HuNoV VLPs were measured using dynamic light scattering (DLS) on a ZetaSizer Nano instrument (Malvern Instruments, UK). Samples were diluted to a final concentration of 330 nM for each component in phosphate-buffered saline pH 6 and a 3,300 nM concentration of NORO-320 Fab was prepared for condition labeled VLP:NORO-320 Fab (1:10). Three × 12 measurement runs were performed with standard settings (Refractive Index 1.335, viscosity 0.9, temperature 25 °C) for each time point. The average result was created with ZetaSizer software.

**Detection of bis-ANS binding by fluorescence spectroscopy.** Purified VLP (30 μg/mL, 0.5 μM concentration of the VP1) or 0.5 μM purified antibody (NORO-320 Fab, IgA, or 22D2 control) diluted in PBS buffer pH 6.0 was incubated at 25 °C or 37 °C to allow for temperature equilibration. To detect bis-(8-anilinonaphthalene-1-sulfonate) (bis-ANS) binding to native VLP and antibody, bis-ANS was added to the sample to a final concentration of 3 μM. Bis-ANS was excited at 395 nm and emission was collected at 495 nm at 30 s intervals for 15 mi on a Flexstation 3 (Molecular Devices, USA). To investigate the binding of bis-ANS to preincubated VLP and antibody, GII.4 VLP and antibody were mixed and incubated for 10 minutes at 25 °C or 37 °C. Bis-ANS then was added to sample, and samples were immediately transferred to a spectrofluorometer for reading as detailed above.

**Virus neutralization assay.** Human jejunal intestinal enteroids (J4$^{Fut2}$ HIEs) were plated and differentiated as cell culture monolayers in collagen IV-coated 96-well plates in commercial Intesticult human organoid growth medium (INT; Stem Cell Technologies), as previously described[52–54]. Prior to infection, 5-fold serial dilutions of NORO-320 IgA, NORO-320 Fab, or a dengue virus-specific control antibody were prepared in CMGF(−) medium supplemented with 500 μM gly-cochenodeoxycholic acid (GCDCA; Sigma, G0759), and each dilution or the medium control was mixed in equal volume with 100 TCID$_{50}$ of GII.4 (GII.P31/GII.4-Sydney/TCH12-580). NORO-320 Fab antibody was also tested to neutralize

**Table 1 Data processing and refinement statistics for GII.4 P-domain–NORO-320 Fab complex.**

| Data collection | |
|---|---|
| Beamline | ALS Beamline 5.0.1 |
| Wavelength, Å | 0.97741 |
| Space group | P21 21 2 |
| Cell dimensions, Å | 119.25, 186.27, 73.44 |
| α,β,γ,° | 90, 90, 90 |
| Resolution, Å | 50–2.25 (2.29–2.25)$^a$ |
| Total reflections | 1716311 |
| Unique reflections | 78053 (3854)$^a$ |
| Redundancy | 6.5 (6.2)$^a$ |
| Completeness (%) | 99.82 |
| <I/sigma> | 15.6875 (2.375)$^a$ |
| $R_{meas}$$^b$ | 0.129 (0.846)$^a$ |
| $R_{pim}$$^b$ | 0.050 (0.340)$^a$ |
| **Refinement statistics** | |
| Resolution, Å | 50–2.25 (2.29–2.25)$^a$ |
| Reflections (work) | 73965 |
| Reflections (test) | 3926 |
| $R_{work}$$^c$ /$R_{free}$$^d$ (%) | 18.08/22.55 |
| *No. of atoms* | |
| Protein | |
| P-domain dimer | 4798 |
| Noro-320 Fab | 6674 |
| Water | 1059 |
| *Average B Value (Å$^2$)* | |
| P-domain dimer | 34.2505 |
| NORO-320 Fab | 31.67 |
| Water | 36.085 |
| **RMSD from ideal geometry** | |
| Bond length (Å) | 0.003 |
| Bond angle (°) | 0.614 |
| **Ramachandran statistics** $^e$ | |
| Favored | 98.38% |
| Outliers | 0.20% |

$^a$ Numbers in parentheses refer to the highest resolution shell.
$^b$ $R_{meas} = \Sigma hkl \{N(hkl)/[N(hkl)-1]\}^{1/2} \times \Sigma_i |I_i(hkl)-\{I(hkl)\}|/\Sigma_{hkl} \Sigma_i I_i(hkl)$ and $R_{pim} = \Sigma_{hkl} (1/(n-1))^{1/2} \Sigma_i |I_{hkl,i} - |/\Sigma_{hkl} \Sigma_i I_{hkl,i}$, where $I_{hkl,i}$ is the scaled intensity of the $i$th measurement of reflection $h, k, l$, is the average intensity for that reflection, and $n$ is the redundancy.
$^c$ $R_{work} = \Sigma_{hkl} |F_o-F_c| / \Sigma_{hkl} |F_o| \times 100$, where $F_o$ and $F_c$ are the observed and calculated structure factors, respectively.
$^d$ Rfree was calculated as for $R_{work}$, but on a test set comprising 5% of the data excluded from refinement.
$^e$ Calculated with MolProbity[48].

GII.17 (GII.P13/GII.17/1295-44). The antibody fragment:virus mixtures were preincubated for 1 h at 37 °C prior to inoculation onto triplicate wells of the differentiated J4$^{Fut2}$ HIE monolayers and incubated for an additional 1 hr at 37 °C. After 1 h post-infection (hpi), monolayers were washed twice with CMGF(-) medium and incubated with differentiation INT medium supplemented with 500 μM GCDCA. After 1 hpi (immediately after wash) and 24 hpi, cells and medium were collected, and RNA was extracted using KingFisher Flex Purification System and MagMax Viral RNA Isolation kit. RNA extracted at 1 hpi was used to determine a baseline value for the amount of input virus that remained associated with cells after washing the inoculated cultures. Virus replication was assessed by quantifying virus genome equivalent levels (GEs) from samples extracted at 24 hpi in comparison to the 1 hpi time point. Percent reduction in GEs relative to medium (100%) then was determined. Reverse-transcription quantitative polymerase chain reaction (RT-qPCR) was performed as described previously[27].

**Reporting summary.** Further information on research design is available in the Nature Research Reporting Summary linked to this article.

## Data availability

Atomic coordinates and structure factors for the crystal structure of the NORO-320 Fab in complex with GII.4 P-domain have been deposited in the Protein Data Bank under the accession code 7JIE. The authors declare that all other data supporting the findings of this study are available within the paper and its supplementary information files. Source data are provided with this paper.

## Materials availability

Materials availability Materials described in this paper are available for distribution for nonprofit use using templated documents from Association of University Technology Managers "Toolkit MTAs", available at: https://autm.net/surveys-and-tools/agreements/material-transfer-agreements/mta-toolkit.

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

## Acknowledgements

The authors thank Nurgun Kose and Robin Bombardi in the Crowe laboratory for excellent technical support. This work was supported by a pilot and feasibility grant from the Vanderbilt University Medical Center's Digestive Disease Research Center supported by NIH grants P30 DK058404 and P01 AI057788 and a grant from the Robert Welch Foundation (Q1279). P30 CA125123 supported the Protein and Monoclonal Antibody Production Shared Resource at Baylor College of Medicine for VLP expression. G.A. was supported through the Vanderbilt Trans-Institutional Program (TIP) "Integrating Structural Biology with Big Data for next Generation Vaccines" and NIH grant F31 AI129357. W.S. was supported through the training fellowship from the Gulf Coast Consortia, on the Training Interdisciplinary Pharmacology Scientists (TIPS) Program (Grant No. T32 GM120011). J.E.C. is a recipient of the 2019 Future Insight Prize from Merck KGaA, which supported this work with a grant. The project described was supported by CTSA award No. UL1 TR002243 from the National Center for Advancing Translational Sciences (NCATS). X-ray diffraction data were collected at the Advanced Light Source (5.0.1) (Berkeley, CA). The contents of this publication are solely the responsibility of the authors and do not necessarily represent the official views of NIGMS, NCATS, NIAID, or NIH.

## Author contributions

G.A., W.S., B.V.V.P., and J.E.C., conceived and designed the research; G.A. isolated and characterized the antibodies. M.K.E. and K.E. provided reagents and performed neutralization assays. W.S., L.H., B.S., and B.V.V.P. determined the X-ray structure of the NORO-320 Fab in complex with the GII.4 P-domain and carried out the structural analyses; G.A., W.S., B.V.V.P., and J.E.C. wrote the manuscript. All authors reviewed, edited, and approved the final manuscript.

## Competing interests

M.K.E. is named as an inventor on patents related to cloning of the Norwalk virus genome, is a consultant to Takeda Vaccines, Inc and has received support from Takeda Vaccines, Inc. J.E.C. has served as a consultant for Luna Biologics, is a member of the Scientific Advisory Boards of Meissa Vaccines and is Founder of IDBiologics. The Crowe laboratory at Vanderbilt University Medical Center has received unrelated sponsored research agreements from Takeda, IDBiologics, and AstraZeneca.
