## [Peer Review File · Nature Communications]

Reviewers' Comments:

Reviewer #1:

Remarks to the Author:

The manuscript by Alvarado, et al., describes the structure of a cross reactive antibody complexed with the P domain of human norovirus and discusses possible neutralization mechanisms. The methods employed are all appropriate and well executed. My major concern is the relative impact of this work since there have been a number of studies already published describing similar, albeit not identical, findings.

The major finding is that the antibody (Fab) is found to bind to the conserved P1 domain. However, this was already noted back in 2012 by Hansman et al. (doi: 10.1128/JVI.06868-11). This was followed by several other papers by Hansman's group using nanobodies (e.g. doi: 10.1128/JVI.02005-18). It is not quite clear how the results presented here represent an advance in the field that merits a Nature publication. These previous publications all showed that the P1 domain is relatively conserved and antibodies that bind to the P1 domain can be highly cross-reactive. Yes, the antibodies studies here are from patients, but that is a relatively minor detail.

Other points that should be addressed:

1) Line 303: I am fairly certain DLS is not sensitive enough to detect significant diameter changes in the particles.

2) I find the discussion of neutralization mechanisms to be not at all clear and a certain amount of caution is merited. Some of the discussion is highly speculative (e.g. line 301 – how would the extracellular antibody block assembly?). There have also been a number of papers on other viruses that extensively discuss various modes of neutralization (e.g. aggregation, steric interference, etc) with the caution that small differences in how the in-vitro assays are performed can yield orders of magnitude differences in apparent neutralization efficacy. The results are often relatively complicated to compare since one has to worry about avidity versus affinity and number of independent particles, etc. Further, all apparent in-vitro neutralization mechanisms may have relatively little impact on in-vivo antibody efficacy as has been shown with several antibody/virus systems.

3) The authors should take more care with citations (e.g. lines 304-314). The flexibility of the Calicivirus P domain was first noted with MNV (doi:10.1128/JVI.02200-07) and then subsequently with RHDV (doi:10.1128/JVI.00314-10) and genotype II (doi: 10.1128/JVI.06868-11). The antibody shown here can only bind to the intact capsid assuming sufficient flexibility in the P domain and therefore these are important initial observations. Finally, the first and most relevant citation for correlation between P domain rotation and increased infectivity should be the work on MNV that demonstrated that bile enhances receptor binding while causing a major rotation and collapse of the P domain onto the shell (DOI: 10.1128/JVI.00970-19).

Reviewer #2:

Remarks to the Author:

Alvarado and colleagues describe the characterization of a panel of cross-reactive human monoclonal antibodies. Particularly, using X-ray crystallography they describe the footprint of a GII-specific antibody, NORO-320, that exhibits HBGA carbohydrate blocking and virus neutralization. Interestingly, NORO-320 Fab did not show HBGA carbohydrate blocking, but neutralized GII.4 infection more potently than the full antibody. Additional data suggest that virus neutralization is mediated by cross-linking of virions by the full antibody, and Fab-mediated neutralization is mediated by a different mechanism. The study is interesting and provides new insights on the antigenic landscape of

noroviruses; however, the authors failed provide insights on the mechanism of neutralization by NORO-320 Fab.

Specific comments:

1. Antibody-mediated blocking of the HBGA carbohydrate interaction with the viral capsid is strongly correlated with virus neutralization (Atmar et al. *J Infect Dis* 2020). Thus, one of the most interesting results of this study is that NORO-320 Fab does neutralize norovirus without blocking the carbohydrate interaction with the viral capsid. Authors discussed the potential mechanisms of Fab-mediated neutralization (Lines 299-314), however, none of the experiments presented offer clues on the mechanism of action. One of the mechanisms discussed is particle conformational changes. Authors could perform biochemical assays (e.g. Ossiboff et al. *J Virol* 2010) to determine minor conformation changes on the VLPs upon binding to the Fab. Moreover, since particle flexibility seems to be required for infection (Song et al. *PLoS Pathog* 2020), the authors could submit VLPs to different conditions (e.g. heat, pH) to determine whether Fab binding influences particle stability. For occluded epitopes like the one described here, differences in the HBGA carbohydrate blocking activity were reported at different temperatures (Lindsmith et al. *mSphere* 2018). Please consider doing the HBGA carbohydrate blocking experiments at 37°C.

2. Potency of viral neutralization by NORO-320 was compared between IgA antibody and Fab (Lines 196-197); however, this is not a fair comparison. The enteroid culture system has variations depending on the virus tested and with each experiment (Constantini et al. *Emerg Infect Dis* 2018; Ford-Siltz et al *J Infect Dis* 2020). The viruses used for infection were different (Alvarado et al. *Gastroenterology* 2018), and the experiment reported in this manuscript was done, in triplicate, once (Figure 5). Thus, comparison of neutralization activity by the two forms of antibody NORO-320 should be tested under the same conditions. Please consider including a control antibody for your experiments.

3. The description of this antibody (NORO-320) opens new opportunities for the development of cross-protective vaccines. However, using high-titer sera from animals immunized with GII.4, GII.6, and GII.17 VLPs, Ford-Siltz et al. (*J Infect Dis* 2020) could not show any cross-neutralization among the genotypes tested. Moreover, multiple instances of reinfection with different genotypes have been reported (Sakon et al. *J Infect Dis* 2015, Dai et al. *J Clin Microbiol* 2017, Karangwa et al. *Open Forum Infect Dis* 2017, Nelson et al. *J Infect Dis* 2018, Chhabra et al. *Clin Infect Dis* 2020), suggesting that cross-reactive neutralizing antibodies are subdominant in the overall immune response. Please discuss the representation of antibodies similar to NORO-320 in a polyclonal response. Did the authors test the sera from the individuals for binding, blocking, or neutralization to different norovirus genotypes?

4. To the best of this reviewer's knowledge, while HBGA carbohydrates have been shown to enhance human norovirus infectivity (Haga et al. *mBio* 2020, Nordgren and Svensson *Viruses* 2019), there is no empirical data showing that they constitute the human norovirus cellular receptor. Please consider to rephrase the term "receptor blocking assay."

5. There are at least ten genogroups in noroviruses (GI-GX). Please update the sentence from lines 77-78 and omit the outdated reference #8.

6. Please consider to update the information and reference from lines 79-80.

7. Please consider to rephrase the statement from lines 99-101. Detailed information on norovirus antigenic diversification was initially collected with mouse monoclonal antibodies, and later confirmed with human sera and monoclonal antibodies (Lindsmith et al. *PLoS Pathog* 2012; Lindsmith et al. *Immunity* 2019). Moreover, rules governing the B-cell immune responses have been shown to be similar in very disparate species (e.g. Altman et al. *eLife* 2015), and monoclonal antibodies from mouse origin are currently being use to control different viral infections.

8. Note that Supplementary Table 1 lists 13 cross-reactive antibodies. Please correct to match that to the manuscript.

9. Please provide data or a reference for the statement "could be due to the broad binding HBGA spectrum of GII.17 HuNoVs."

10. Please consider to remove reference #28 (line 159) and use reference #8 instead.

11. The proof that NORO-320 neutralizes norovirus is supported by experiments using the enteroids culture system. Please consider to rephrase the statement from lines 174-176.

12. Please consider to rephrase the statements from lines 247-250.

Reviewer #3:

Remarks to the Author:

The authors describe the evaluation of a panel of 12 human monoclonal antibodies against human norovirus which, with a panel of different norovirus VLPs, they were able to group into three different binding patterns. One of these IgA MoAbs, NORO-320, was previously shown to exhibit receptor blockade and inhibit replication of infectious GII.4 Sydney in a human intestinal enteroid culture. Interestingly, this MoAb blocks GII.4 and GII.6 VLPs binding to porcine gastric mucine, but not to GII.17. The then elegantly show that Noro_320 Fab did not block GII.4 Sydney VLPs from binding to PGM and that the 320Fabs neutralizes infectious GII.4 and GII.17 with a lower IC50 than complete NORO 320 IgA suggesting a neutralization mechanism different than HBGA blockade which is one of the major finding of this paper. The latter was confirmed by crystal structure analysis with GII.4 P domain.

This manuscript is well written and provides convincing evidence of another mechanism than the currently accepted of HBGA blocking, of neutralization of norovirus directed against the P-domain of the capsid. Human noroviruses have long been poorly studied while they have a tremendous burden (disease and societal cost) worldwide. Recent development of a human intestinal cell culture have allowed a better studying of this group of heterogeneous RNA viruses.

Then the authors argue that the reason why the GII VLPs all bind to NORO-320 is because they have 78-89% sequence alignment. However, most genotypes differ from each other at least 20% and therefore 78% sequence identity is quite significant. Please discuss. Also, to generalize statements based on a single VLPs representing an entire genotype should be mentioned as a limitation as significant sequence variation within a genotype exist

Specific comments

Line 36 delete NoV

Line 47 which MAbs?

Line 71 replace 'NoVs' with noroviruses

Line 72 NoV = norovirus

Line 77 NoVs = noroviruses

Line 78 7 different (Vinje 2015) = 10 different genogroups (G) (in updated classification by Chhabra et al 2019); 'The viruses that infect humans' = 'HuNoVs'. Delete 'genogroup I, II, IV, VIII, and IX'

Line 79 remove bracket, 'are' = 'can be'

Line 87-89 Please make more clear which MAbs were already reported in previous publications (references 22 and 23). In line 119, 5 MAbs were previously characterized. Also in ref 22 PBMCs from Norwalk virus (GI.1) infected volunteers were isolated. Were MoAbs derived from these individuals reacting only against GI VLPs?

Line 105 'These'? 'Our' perhaps?

RESULTS Overall can be condensed to take out M&M language (e.g., 139-142) to the M&Ms and just report the results. Differences in EC50 (Figure 3 are not well discussed)

FIGURES

Fig 1B graphs are pretty small and curves of NORO 320 which is the most interesting in the paper should be highlight in a better way and get easy lost with buried by all the other ELISA binding curves.

Fig 9 line numbers are mixed up with title of Fig 9B

Supplemental Table 1. The paper describes 12 MABs but this table listed antibody sequence of 13 (including 161.2). What was the reason to leave this antibody out of the main paper?

Nature Communications manuscript NCOMMS-20-48256
RESPONSE TO REVIEWER COMMENTS

Reviewer #1

The manuscript by Alvarado, et al., describes the structure of a cross reactive antibody complexed with the P domain of human norovirus and discusses possible neutralization mechanisms. The methods employed are all appropriate and well executed. My major concern is the relative impact of this work since there have been a number of studies already published describing similar, albeit not identical, findings.

We are not aware of any other human antibodies derived from a natural infection described to date with these features, and thus we view the findings as new and informative for the field. The antibodies described by Baric and collaborators are GII.4-specific, whereas the antibodies described in the studies presented here are broadly cross-reactive among GII genogroup viruses. Moreover, we are not aware of other studies describing the potent neutralization activity of the Fab antibody fragment alone, as we have described here. The epitopes on the P domain that we have identified are also unique and conserved in most GII HuNoVs, providing a basis for the broad cross-reactivity of our antibody.

The major finding is that the antibody (Fab) is found to bind to the conserved P1 domain. However, this was already noted back in 2012 by Hansman et al. (doi: 10.1128/JVI.06868-11). This was followed by several other papers by Hansman's group using nanobodies (e.g. doi: 10.1128/JVI.02005-18). It is not quite clear how the results presented here represent an advance in the field that merits a Nature publication. These previous publications all showed that the P1 domain is relatively conserved and antibodies that bind to the P1 domain can be highly cross-reactive. Yes, the antibodies studies here are from patients, but that is a relatively minor detail.

We have to respectfully disagree with the reviewer on this point. While we do not discount the importance of the previous work in model systems, many investigators also would agree that it is important and significant to determine the patterns of recognition of naturally occurring human B cells reacting to human pathogens in order to understand the human immunobiology properly. These referenced papers did identify structural features recognized by one mouse mAb (5B18) and three single chain Fvs from one alpaca. However, it is well established that murine antibody repertoires differ significantly from human repertoires in terms of size, complexity, length of HCDR3s and other important features. Camelid antibodies use completely different binding modes from human antibodies formed by heavy and light chains.

1) Line 303: I am fairly certain DLS is not sensitive enough to detect significant diameter changes in the particles.

DLS is sensitive to detect changes when the changes in the diameter are significant. This technique can distinguish between small particles, completely disassembled particles, and

larger particle aggregates from the normal sized VLPs. However, DLS is not sensitive enough to detect smaller conformational changes (for example, resting and raised conformations of the P domain in the VLPs, as has been suspected to occur in some noroviruses). Also, note that DLS has been used previously by Hansman's group (Koromyslova *et al.*, *PLOS Path.* 2017) to clearly show significant changes in particle diameter when treated with nanobodies. This previous work showed that DLS is sensitive enough to detect significant changes in particle diameter.

Updated revision lines 319-326 was edited to clarify: The DLS data presented here indicate that binding of Fab-320 does not compromise particle integrity, because the diameter of the VLPs remains same as that of the VLPs in the absence of the Fab-320 (40 nm). In contrast, we observe a significant increase from 40 nm to 210 nm with IgA, strongly suggestive of particle aggregation due to multivalent cross-linking, or aggregation of the sub-assemblies following particle disassembly. We have substantiated this finding further by performing the DLS experiments with an irrelevant antibody, which showed no difference in the VLP diameter.

2) I find the discussion of neutralization mechanisms to be not at all clear and a certain amount of caution is merited. Some of the discussion is highly speculative (e.g., line 301 – how would the extracellular antibody block assembly?). There have also been a number of papers on other viruses that extensively discuss various modes of neutralization (e.g., aggregation, steric interference, etc.) with the caution that small differences in how the in-vitro assays are performed can yield orders of magnitude differences in apparent neutralization efficacy. The results are often relatively complicated to compare since one has to worry about avidity versus affinity and number of independent particles, etc. Further, all apparent in-vitro neutralization mechanisms may have relatively little impact on in-vivo antibody efficacy as has been shown with several antibody/virus systems.

We apologize for the mistake, we meant disassembly not assembly (**revision line 316**). We have edited the **revision lines 313-327** to provide caveats about mechanistic interpretation, as suggested. The mechanism of neutralization by NORO-320 IgA or IgG likely involves aggregation and/or particle disassembly as suggested by our DLS experiments. It is possible that either of these two processes would severely affect HBGA binding as shown by our HBGA blocking assays (**Figure 4**). However, in contrast with the NORO-320 Fab that effectively neutralizes the infectivity, we did not observe either the blocking of HBGA binding or any disruption of the particle integrity. Furthermore, our experiments with bis-ANS (see below) also showed that binding of NORO-320 Fab does not cause any significant conformational changes. These observations led us to hypothesize that mechanism of neutralization by the NORO-320 Fab is by preventing the conformational changes that are obligatory during the cell entry process, for example, for binding to a yet-to-be-discovered proteinaceous receptor.

3) The authors should take more care with citations (e.g. lines 304-314). The flexibility of the Calicivirus P domain was first noted with MNV (doi:10.1128/JVI.02200-07) and then subsequently with RHDV (doi:10.1128/JVI.00314-10) and genotype II (doi: 10.1128/JVI.06868-11). The antibody shown here can only bind to the intact capsid assuming sufficient flexibility in the P domain and therefore these are important initial observations. Finally, the first and

most relevant citation for correlation between P domain rotation and increased infectivity should be the work on MNV that demonstrated that bile enhances receptor binding while causing a major rotation and collapse of the P domain onto the shell (DOI: 10.1128/JVI.00970-19).

Thank you. We have updated the references accordingly for **revision lines: 332-337**.

Reviewer #2

1. Antibody-mediated blocking of the HBGA carbohydrate interaction with the viral capsid is strongly correlated with virus neutralization (Atmar et al. J Infect Dis 2020). Thus, one of the most interesting results of this study is that NORO-320 Fab does neutralize norovirus without blocking the carbohydrate interaction with the viral capsid. Authors discussed the potential mechanisms of Fab-mediated neutralization (Lines 299-314), however, none of the experiments presented offer clues on the mechanism of action. One of the mechanisms discussed is particle conformational changes. Authors could perform biochemical assays (e.g. Ossiboff et al. J Virol 2010) to determine minor conformation changes on the VLPs upon binding to the Fab.

Based on the suggestion of reviewer #2, we performed the biochemical assay discussed in Ossiboff *et al.* (*J Virol* 2010). We used binding of bis-ANS as a probe to investigate changes in hydrophobicity that occurred upon interaction of GII.4 VLP with NORO-320 Fab/IgA in solution. Bis-ANS is a fluorophore that binds to exposed hydrophobic regions of protein molecules, and the sequestration of bis-ANS in hydrophobic pockets of proteins is accompanied by a large increase in the fluorescence quantum yield of the probe. Thus, bis-ANS can be used to probe for structural changes in viral proteins that are accompanied by changes in surface hydrophobicity. Shown in **Supplemental Figure 3**, NORO-320 Fab and IgA do not induce an increase in fluorescence upon binding to GII.4 VLP. These data suggest that NORO-320 Fab alone does not induce minor conformational changes in the VLP. This finding suggests that possible modes of neutralization are that NORO-320 may interfere with the conformational changes induced upon cellular receptor engagement or it disrupts the binding of co-receptors involved in cellular entry.

The bis-ANS results are shown in Supplemental Figure 3 and revision lines 243-250.

Moreover, since particle flexibility seems to be required for infection (Song et al. PLoS Pathog 2020), the authors could submit VLPs to different conditions (e.g. heat, pH) to determine whether Fab binding influences particle stability.

To test for Fab binding influencing particle stability, we performed DLS experiments at a range of temperature (25°C to 40°C) and pH (6.0 to 8.0) conditions. The addition of NORO-320 Fab does not induce the additional formation of smaller particles (~ 20 nm diameter) or aggregation of particles at any of the temperatures tested.

The results are shown in Supplemental Figure 4 and *revision* lines 239-241.

For occluded epitopes like the one described here, differences in the HBGA carbohydrate blocking activity were reported at different temperatures (Lindesmith et al. mSphere 2018). Please consider doing the HBGA carbohydrate blocking experiments at 37°C.

In light of the findings from the additional experiments that we performed regarding the possible neutralization mechanism by IgA or the Fab as discussed above, we consider these additional HBGA blocking experiments less likely to yield helpful information and beyond the scope of this manuscript.

2. Potency of viral neutralization by NORO-320 was compared between IgA antibody and Fab (Lines 196-197); however, this is not a fair comparison. The enteroid culture system has variations depending on the virus tested and with each experiment (Constantini et al. Emerg Infect Dis 2018; Ford-Siltz et al J Infect Dis 2020). The viruses used for infection were different (Alvarado et al. Gastroenterology 2018), and the experiment reported in this manuscript was done, in triplicate, once (Figure 5). Thus, comparison of neutralization activity by the two forms of antibody NORO-320 should be tested under the same conditions. Please consider including a control antibody for your experiments.

For consistency and reproducibility of results, we repeated the neutralization assays using the GII.4/TCH12-580 strain that was used in previous work (Alvarado et al., Gastroenterology 2018) using NORO-320 IgA or Fab under the same conditions. A dengue virus-specific antibody was included as a control antibody. **All neutralization assays were performed more than once, and data from a representative experiment are shown in the modified Figure 5.**

3. The description of this antibody (NORO-320) opens new opportunities for the development of cross-protective vaccines. However, using high-titer sera from animals immunized with GII.4, GII.6, and GII.17 VLPs, Ford-Siltz et al. (J Infect Dis 2020) could not show any cross-neutralization among the genotypes tested. Moreover, multiple instances of reinfection with different genotypes have been reported (Sakon et al. J Infect Dis 2015, Dai et al. J Clin Microbiol 2017, Karangwa et al. Open Forum Infect Dis 2017, Nelson et al. J Infect Dis 2018, Chhabra et al. Clin Infect Dis 2020), suggesting that cross-reactive neutralizing antibodies are subdominant in the overall immune response. Please discuss the representation of antibodies similar to NORO-320 in a polyclonal response. Did the authors test the sera from the individuals for binding, blocking, or neutralization to different norovirus genotypes?

It is commonly observed in animal and human studies of viral immunology that individual mAbs with a high degree of breadth and potency can be isolated from the memory B cells of individuals in whom the serum antibodies (secreted by long lived plasma cells in the bone marrow) are not appreciable. In fact, this is a major focus of studies in the area of reverse

vaccinology studies for HIV, influenza, hepatitis virus and others. We agree with the reviewer that the clear observation for noroviruses is that the type of antibody we describe here is subdominant in the natural response and indicate this in the revised manuscript to highlight the reviewer's insight on this point.

4. To the best of this reviewer's knowledge, while HBGA carbohydrates have been shown to enhance human norovirus infectivity (Haga et al. mBio 2020, Nordgren and Svensson Viruses 2019), there is no empirical data showing that they constitute the human norovirus cellular receptor. Please consider to rephrase the term "receptor blocking assay."

We agree we should change this to HBGA-blocking assay. We rephrased the term in the main text.

5. There are at least ten genogroups in noroviruses (GI-GX). Please update the sentence from lines 77-78 and omit the outdated reference #8.

Thank you. The reference and genogroup information has been updated in *revision lines 78-79*.

6. Please consider to update the information and reference from lines 79-80.

Thank you. The information and references have been updated from *lines 79-80*.

7. Please consider to rephrase the statement from lines 99-101. Detailed information on norovirus antigenic diversification was initially collected with mouse monoclonal antibodies, and later confirmed with human sera and monoclonal antibodies (Lindesmith et al. PLoS Pathog 2012; Lindesmith et al. Immunity 2019). Moreover, rules governing the B-cell immune responses have been shown to be similar in very disparate species (e.g. Altman et al. eLife 2015), and monoclonal antibodies from mouse origin are currently being use to control different viral infections.

We considered this point. Clearly the overall general immunologic principles of B cell activation and antibody actions are similar in many mammals. However, we don't think it is scientifically correct to assume animal and human antibody responses coincide at the genetic, molecular, and structural level. Murine variable gene sequences differ from human and murine CDRs tend to be shorter and less complex than human CDRs. The immunodominance patterns for epitope recognition by murine mAbs differs greatly from that of humans for many virus systems. Human-type responses for broad neutralizing Abs to HIV envelope cannot be recapitulated in mice, rabbits, or even macaques. Murine responses to flaviviruses focus on Domain III of the Envelope protein, whereas human responses focus more commonly on DII and complex quaternary epitopes. The similarities in murine and human responses tend to break down at the level of fine epitope recognition, which is the focus of this paper.

We also are not aware of any murine mAbs in clinical use to control virus infections. Palivizumab is the single legacy humanized murine mAb for a virus that is licensed and used

clinically, but it is in the process of being replaced by the sponsor AZ with a fully human mAb, which is preferred.

8. Note that Supplementary Table 1 lists 13 cross-reactive antibodies. Please correct to match that to the manuscript.

Thank you, we have corrected this point by removing one antibody listing from **Supplemental Table 1** (as this clone expressed very poorly, and we were unable to study its characteristics). The number 12 **on revision line 128** is correct).

9. Please provide data or a reference for the statement "could be due to the binding HBGA spectrum of GII.17 HuNoVs."

We agree this point was unclear. We clarified this point in the revised manuscript. Here we are trying to provide a reason as to why NORO-320 IgA neutralizes effectively but does not block HBGA binding in the case of GII.17. This finding is certainly intriguing. One possibility is that our HBGA blocking assay with pig gastric mucin does not entirely recapitulate the cellular glycans to which GII.17 viruses binds. Another possibility is that the HBGA binding site remains available despite the aggregation or disassembly that NORO-320 IgA is suggested to cause. **We updated revision lines 149-153 based on this comment.**

10. Please consider to remove reference #28 (line 159) and use reference #8 instead.

This comment is a little unclear. The line 159 does not cite ref 28. However, ref 28 is cited in line 155, and this citation seem appropriate.

11. The proof that NORO-320 neutralizes norovirus is supported by experiments using the enteroids culture system. Please consider to rephrase the statement from lines 174-176.

We have updated the text in **revision lines 175-176.**

12. Please consider to rephrase the statements from lines 247-250.

We have updated the text for **revision lines 255-257.**

Reviewer #3

Then the authors argue that the reason why the GII VLPs all bind to NORO-320 is because they have 78-89% sequence alignment. However, most genotypes differ from each other at least 20% and therefore 78% sequence identify is quite significant. Please discuss.

Thank you, we edited the text at **revision lines 226-230** to point this out. When the entire P-domain sequence is aligned between the Genogroup II strains, the overall sequence alignment

is 54-59%. Thus, the NORO-320 epitope has a higher degree of sequence conservation compared to the entire P-domain sequence.

Also, to generalize statements based on a single VLPs representing an entire genotype should be mentioned as a limitation as significant sequence variation within a genotype exist

Thank you, we added text to highlight this limitation in the Discussion, **revision lines 280-281**.

Line 36 delete NoV

We edited as suggested.

Line 47 which MAbs?

We edited to clarify NORO-320.

Line 71 replace 'NoVs' with noroviruses

We edited as suggested.

Line 72 NoV = norovirus

We edited as suggested.

Line 77 NoVs = noroviruses

We edited as suggested.

Line 78 7 different (Vinje 2015) = 10 different genogroups (G) (in updated classification by Chhabra et al 2019); 'The viruses that infect humans' = 'HuNoVs'. Delete 'genogroup I, II, IV, VIII, and IX'

We edited as suggested.

Line 79 remove bracket, 'are' = 'can be'

We edited as suggested.

Line 87-89 Please make more clear which MAbs were already reported in previous publications (references 22 and 23).

We edited to clarify in **revision lines 89-90**.

In line 119, 5 MAbs were previously characterized. Also in ref 22 PBMCs from Norwalk virus (GI.1) infected volunteers were isolated. Were MoAbs derived from these individuals reacting only against GI VLPs?

We edited the text to clarify in *revision lines 90-91*.

The ten mAbs from reference 22 were GI.1-specific, and they are not studied in the current paper.

Five of the 25 GII.4-reactive mAbs from reference 15 are included in this paper. The previous studies did not address the cross-reactivity of these clones, which was discovered later as part of the studies in the current paper.

Line 105 'These'? 'Our' perhaps?

We edited as suggested.

RESULTS Overall can be condensed to take out M&M language (e.g., 139-142) to the M&Ms and just report the results.

We edited the results section as suggested in multiple sections of the Results portion of the manuscript.

Differences in EC50 (Figure 3 are not well discussed)

We added discussion as suggested, remarking on the ranges and specifically the values for NORO-320.

FIGURES

Fig 1B graphs are pretty small and curves of NORO 320 which is the most interesting in the paper should be highlight in a better way and get easy lost with buried by all the other ELISA binding curves.

We updated the figure to increase the size of the plots, and we highlighted the curve for NORO-320 with text indicators.

Fig 9 line numbers are mixed up with title of Fig 9B

We updated the document.

Supplemental Table 1. The paper describes 12 MAbs but this table listed antibody sequence of 13 (including 161.2). What was the reason to leave this antibody out of the main paper?

Thank you for picking up on this. That antibody NORO-161.2 expressed very poorly, and we were unable to study its characteristics. We have removed it from the Supplemental Table 1.

Reviewers' Comments:

Reviewer #2:

Remarks to the Author:

This reviewer acknowledges the changes included in the revised version of this manuscript and finds the data presented very interesting. However, I still believe that the authors failed to provide sufficient experimental information to support their claims that the Fab version of NORO-320 neutralizes more potently than the mAb, and for the mechanisms of neutralization by this antibody clone. Thus, I agree with other reviewers that caution is warranted on the proposed mechanisms of neutralization.

1. Authors' suggest that carbohydrate blocking and neutralization of the virus by mAb is due to multivalent cross-linking, but this reviewer is not quite sure why aggregation would completely prevent carbohydrate binding, as those regions involved in HBGA interaction would still be accessible on aggregated particles. To prove this, authors could provide evidence that other mAbs that do not block carbohydrate interaction (Figures 1 and 2), do not cause aggregation of VLPs.

2. Authors' suggest that "NORO-320 Fab has ~ 1.3-fold stronger neutralization potency than NORO-320 IgA", however this difference is minimal and to support this claim, statistical analyses should be presented.

Regarding the variability of these experiments, the authors replied "All neutralization assays were performed more than once, and data from a representative experiment are shown in the modified Figure 5." The data for genome copies is normalized to the control experiment, so there is no reason to not include all data collected in one single figure. This additional data could provide enough power for the statistical analyses to support (or not) differences in potency.

3. Authors' used the biochemical analyses (DLS, bis-ANS) to speculate that "NORO-320 Fab neutralizes the virus by preventing the conformational changes that are obligatory during the cell entry process, for example, for binding to a yet-to-be-discovered proteinaceous receptor." The conformation changes reported for MNV include a rising and resting state of the P domain (Song et al. Plos Pathog 2020), which might not result in substantial changes in surface accessibility of the original conformation. Therefore, this reviewer has reservations that the data provided could be used to propose they found a new mechanism of neutralization; further research is necessary in that regard. Again, some caution is advised on claims that a novel mechanism of neutralization for noroviruses has been identified, as this could well be a case that the "proteinaceous" human receptor binds close the epitope of NORO-320 and distal from the HBGA binding sites.

Minor changes:

Lines 95: Please note the typo on "blocking"

Lines 104-105: Please consider to revise the grammar and writing style of the following phrase: "Therefore, we set out to isolate HuNoV cross-reactive broadly blocking A histo-blood group antigen mAbs and to map their corresponding epitopes"

Lines 108-109: Until further information is included, I suggest removing "We also identify here a novel mechanism of human antibody-mediated HuNoV neutralization."

Lines 147-148: I apologize that I did not notice this earlier, but the authors stated that "None of the 8 mAbs with binding reactivity to GII.17 VLPs had any GII.17 blocking activity", however in Figure 2, Panel B, it is clearly noticeable that two antibodies do have, albeit limited, blocking activity. Please comment on how EC50 values were calculated.

Line 285: Please consider the use of "viral particle" instead of "live virion".

Line 552: Please provide a reference for the ELISA assay previously described.

Response to reviewer comments

Reviewer #2

This reviewer acknowledges the changes included in the revised version of this manuscript and finds the data presented very interesting. However, I still believe that the authors failed to provide sufficient experimental information to support their claims that the Fab version of NORO-320 neutralizes more potently than the mAb, and for the mechanisms of neutralization by this antibody clone. Thus, I agree with other reviewers that caution is warranted on the proposed mechanisms of neutralization.

We edited the manuscript throughout to remove explicit claims of two separate mechanisms of neutralization, or discovery of novel modes of neutralization, as detailed in responses below to particular passages of text. We also edited the statement in the abstract to soften this point:

“Aggregation was not observed with the Fab form of NORO-320, suggesting that this clone also has additional inhibitory features.”

1. Authors' suggest that carbohydrate blocking and neutralization of the virus by mAb is due to multivalent cross-linking, but this reviewer is not quite sure why aggregation would completely prevent carbohydrate binding, as those regions involved in HBGA interaction would still be accessible on aggregated particles. To prove this, authors could provide evidence that other mAbs that do not block carbohydrate interaction (Figures 1 and 2), do not cause aggregation of VLPs.

We agree that the mechanisms of aggregation and HBGA blocking would not necessarily have to be directly linked in all cases for all antibodies. We altered the text on this matter to better stick to the data observed and removed the explicit language around two distinct mechanisms:

“Based on these observations, we suggest that NORO-320 IgA likely mediates neutralization principally by particle aggregation or disassembly of GII.4 particles. Since the Fab form of NORO-320 also mediates neutralization without causing aggregation, there must also be additional inhibitory features of this antibody that we did not define here that contribute to neutralization.”

2. Authors' suggest that "NORO-320 Fab has ~ 1.3-fold stronger neutralization potency than NORO-320 IgA", however this difference is minimal and to support this claim, statistical analyses should be presented.

We agree this difference in potency is small. We have removed the statement on the difference in potency, and the corresponding statement about the suggestion of a non-HBGA blocking mechanism.

Regarding the variability of these experiments, the authors replied "All neutralization assays were performed more than once, and data from a representative experiment are shown in the modified Figure 5." The data for genome copies is normalized to the control experiment, so

there is no reason to not include all data collected in one single figure. This additional data could provide enough power for the statistical analyses to support (or not) differences in potency.

We now include all data in a single figure, revised Figure 5. Statistical analysis was performed using Student's t test.

3. Authors' used the biochemical analyses (DLS, bis-ANS) to speculate that "NORO-320 Fab neutralizes the virus by preventing the conformational changes that are obligatory during the cell entry process, for example, for binding to a yet-to-be-discovered proteinaceous receptor." The conformation changes reported for MNV include a rising and resting state of the P domain (Song et al. Plos Pathog 2020), which might not result in substantial changes in surface accessibility of the original conformation. Therefore, this reviewer has reservations that the data provided could be used to propose they found a new mechanism of neutralization; further research is necessary in that regard. Again, some caution is advised on claims that a novel mechanism of neutralization for noroviruses has been identified, as this could well be a case that the "proteinaceous" human receptor binds close the epitope of NORO-320 and distal from the HBGA binding sites.

We removed statements about a proteinaceous receptor. We edited the text discussing the effect the Fab to avoid an over-reaching interpretation:

“The binding of NORO-320 Fab possibly could play a role in impeding such rotation and elevation between conformations, resulting in lower infection efficiency, although we did not demonstrate this effect directly. Such an effect of this Fab could pertain to GII.3, GII.6, and GII.17 viruses, since the amino acids critical for the interaction between the GII.4 P-domain and NORO-320 are conserved in those strains.”

Minor changes:

Lines 95: Please note the typo on "blocking"

Thank you, we have now corrected this typo.

Lines 104-105: Please consider to revise the grammar and writing style of the following phrase: "Therefore, we set out to isolate HuNoV cross-reactive broadly blocking A histo-blood group antigen mAbs and to map their corresponding epitopes"

Thank you, we have now revised the text for clarity:

“Therefore, we set out to isolate HuNoV-specific human mAbs with histo-blood group antigen blocking activity that cross-react with diverse strains and then to map their epitopes.”

Lines 108-109: Until further information is included, I suggest removing "We also identify here a novel mechanism of human antibody-mediated HuNoV neutralization."

We removed this statement as suggested.

Lines 147-148: I apologize that I did not notice this earlier, but the authors stated that "None of the 8 mAbs with binding reactivity to GII.17 VLPs had any GII.17 blocking activity", however in Figure 2, Panel B, it is clearly noticeable that two antibodies do have, albeit limited, blocking activity. Please comment on how EC50 values were calculated.

We added a statement about how EC₅₀ values were calculated to the Figure 2 legend.

We revised the text to note that two mAbs appeared to exhibit a low level of activity against GII.17:

“None of the 8 mAbs with binding reactivity to GII.17 VLPs had any strong blocking activity with GII.17; two clones exhibited some activity, but the EC₅₀ values were estimated to be > 1,000 nm.”

Line 285: Please consider the use of "viral particle" instead of "live virion".

We changed the text as suggested.

Line 552: Please provide a reference for the ELISA assay previously described.

We now provide a reference for the ELISA.